

# Two-dimensional Ising and Potts model with long-range bond disorder: A renormalization group approach

Francesco Chippari[1⋆], Marco Picco[1†] and Raoul Santachiara[2‡]

**1** Sorbonne Université & CNRS, UMR 7589, LPTHE, F-75005, Paris, France
**2** Université Paris-Sud, Laboratoire de Physique Théorique et Modèles Statistiques

⋆ fchippari@lpthe.jussieu.fr , † picco@lpthe.jussieu.fr , ‡ raoul.santachiara@u-psud.fr

## Abstract

In this paper we provide new analytic results on two-dimensional $q$-Potts models ($q \geq 2$) in the presence of bond disorder correlations which decay algebraically with distance with exponent $a$. In particular, our results are valid for the long-range bond disordered Ising model ($q = 2$). We implement a renormalization group perturbative approach based on conformal perturbation theory. We extend to the long-range case the RG scheme used in [V. Dotsenko et al., Nucl. Phys. B 455 701-23] for the short-range disorder. Our approach is based on a 2-loop order double expansion in the positive parameters $(2-a)$ and $(q-2)$. We will show that the Weinrib-Halperin conjecture for the long-range thermal exponent can be violated for a non-Gaussian disorder. We compute the central charges of the long-range fixed points finding a very good agreement with numerical measurements.

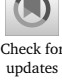

# 1  Introduction

We study here the two-dimensional $q$−Potts model with a long-range correlated bond disorder. On the basis of Monte Carlo results, we conjectured in [1] the phase diagram shown in Fig. (1). Depending on the value of the number of spin's states $q$ and of the disorder power-law decaying exponent $a$, defined in Eq.(2) below, the critical behavior of the system is determined by four critical points: the pure (P) point, which describes the system without disorder, the SR point, which is dominated by the short-range part of the disorder distribution, and the LR and LRp points, where instead the disorder long-range nature prevails. The LR point is associated to a finite strength of the disorder while the LRp is an infinite disorder one. We refer the reader to [1], and references therein, for a more general introduction to long-range disorder systems. Long-range disordered models were studied using a renormalization group (RG) approach long ago [2–7]. These RG computations are based on approximations which are valid in the proximity of the upper critical dimension, $d \sim d_c$, for instance $d_c = 4$ for the Ising model [2,3,5] or $d_c = 6$ for the pure long-range percolation [4], and for small values of $4-a$. At the perturbation order at which the RG computations were done, typically 1-loop order, a stable LR fixed point appeared and the corresponding thermal exponent $\nu^{LR}$ was found to be $\nu^{LR} = 2/a$: this is the so called Weinrib-Halperin conjecture [2]. In [6,7] a 2-loop order RG analysis was implemented to study the $d = 3$ long-range bond disordered Ising for $2 < a < 3$. A quantitatively important violation of the Weinrib-Halperin conjecture, difficult to ascribe to the approximation scheme, was found. The Monte Carlo simulations are not conclusive about this point [8–11].

Much less analytic results are available in $d = 2$ dimension and concern only the long-range disordered Ising model [12,13] which has the nice property of being represented by a free fermion field theory [14]. In particular, a 2-loop order RG computation based on a double expansion on $2 - a$ and $2 - d$ was carried out in [13], supporting in this case the Weinrib-Halperin conjecture.

In this paper we implement a RG perturbative approach based on conformal perturbation theory, by extending to the long-range case the RG scheme used in [15]. This allows us to provide new analytic results valid on the so far unexplored region $q \geq 2$ and $a < 2$, with $a-2 \ll 1$, $q-2 \ll 1$, $(q-2)/(a-2)$ finite. To test our theory, we compute the central charge

of the long-range Ising and Potts model that will be compared to numerical transfer matrix results. The long-range correlation length exponent $\nu^{SR}$ will be computed as well at 2-loop order.

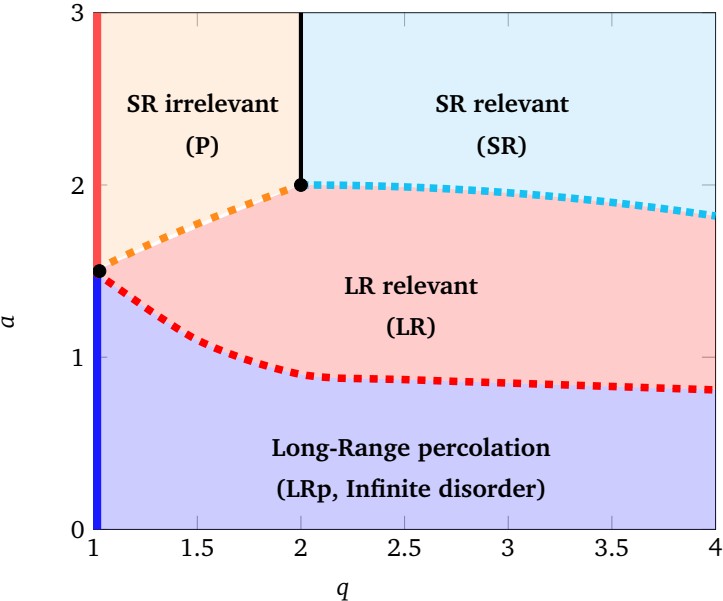

Figure 1: The Figure shows the phase diagram, proposed in [1], of the two-dimensional $q$−Potts model with long-range bond disorder. The parameter $a$, on the vertical axis, is the disorder power-law decaying exponent, defined in Eq. (2). The solid black, dashed cyan, dashed orange and dashed red lines indicate respectively the exchange of stability between the P-SR, the SR-LR, the P-LR and the LR-LRp fixed points. The P-SR and SR-LR lines are conjectured to be described respectively by the equations $a = 2/\nu^P(q)$ and $a = 2/\nu^{SR}(q)$. The SR-LR line is derived in Eq. (26) and in Eq. (34). The exact location of LR-LRp line is beyond reach for our RG expansion, still we can predict the existence of such line, see the Eq. (31) and the Eq. (34).

It is important to stress that the theoretical literature so far considered only Gaussian disorder distributions. However, contrary to the short-range disorder, the terms generated by the higher cumulants can be relevant and can therefore change some universal observable. The phenomena has been pointed out in [1]. At the best of our knowledge, the role of higher cumulants has never been satisfactory explored so far. In this paper we show that the Weinrib-Halperin conjecture is satisfied for a Gaussian disorder and is violated for an instance of a non-Gaussian disorder distribution.

## 2 The effective field theory in the replica approach

In this section we will introduce the effective field theory which describes the continuum limit of the Potts model with long-range bond disorder. For an explicit lattice representation of this model we refer the reader to the Section (2) of [1].

## 2.1 Disorder field

In nature, long-range disorder appears in systems where the impurities form spatially corre­lated structures. Examples are lines of non-magnetic defects, see [8] and references therein, or porous media through which a quantum fluid is transported [16,17]. In general, in the exper­imental realizations, the disorder distribution is not Gaussian and only its first two moments are known.

We consider here disorder distributions whose measure admits a quantum field theory formulation. This formulation allows a precise definition of the disorder cumulants on the one side, and it can be applied to most of the disorder distributions considered in theoretical and numerical works, on the other.

We define a disorder field, which we denote as the $\sigma(x)$ field whose statistical properties are encoded in a functional $\mathcal{S}^{\text{aux}}[\phi(x)]$ of a field $\phi(x)$. The disorder field $\sigma(x) = \sigma[\phi(x)]$ will be some function of $\phi(x)$. The averages over disorder, denoted here as $\mathbb{E}[\cdots]$, take in this setting the form of a path integral:

$$\mathbb{E}[\cdots] = \frac{1}{\int \mathcal{D}\phi\, e^{\mathcal{S}^{\text{aux}}[\phi]}} \int \mathcal{D}\phi\, e^{\mathcal{S}^{\text{aux}}[\phi]} \ldots \tag{1}$$

In general, we will not need to specify the dependence on $\phi$ neither of the action $\mathcal{S}^{\text{aux}}[\phi]$, nor of the field $\sigma[\phi]$. Indeed, in a conformal bootstrap approach, once assumed $\mathcal{S}^{\text{aux}}$ to be a (global) conformal action, the expectation values are computed without using the path integral formulation [18]. A long-range correlated disorder distribution is characterized by the following properties:

$$\mathbb{E}[\sigma] = 0\,, \qquad \mathbb{E}[\sigma(x)\sigma(y)] = |x-y|^{-a}\,, \qquad \mathbb{E}[\sigma(x)\sigma(y)\sigma(z)] = 0\,. \tag{2}$$

The disorder is long-range correlated with a decay exponent $a$. In CFT bootstrap jargon, see [18], the above assumptions (2) can be reformulated by saying that $\sigma(x)$ is a primary field with scaling dimension:

$$h_\sigma = \frac{a}{2}\,, \tag{3}$$

and that the operator product expansion (OPE) $\sigma \times \sigma$ does not produce another $\sigma$ field.

Theoretical works [2–7,12,13] considered the scale-invariant Gaussian disorder distribu­tion defined in Appendix (D). Numerical implementations of a Gaussian disorder can be found in [1,8,10,11]. Some numerical works considered non-Gaussian disorder, for instance in [8], which are not described by any known scale invariant field theory. Other numerical works generated instead non-Gaussian long-range disorder by taking $\sigma$ as the scaling observable of a critical lattice model, see [1,19,20]. In [19] a diluted $d = 3$ Ising model is considered where the vacancies are placed at the locations of the minority spins of an auxiliary pure $d = 3$ Ising model at criticality. The $\mathcal{S}^{\text{aux}}$ is therefore the action describing the conformal critical point of the $d = 3$ Ising model. In [1,20] $d = 2$ disordered Potts model were considered. In [20], the disorder field $\sigma$ coincides with the polarization density field of a $d = 2$ Ashkin-Teller (AT) model. In this case the $\mathcal{S}^{\text{aux}}$ is the action which describes the AT critical line, which, in turn, can be expressed in term of a compactified (and orbifolded) free scalar field [21]. In [1], $\sigma(x) = \prod_i^m \sigma_i(x)$, where $\sigma_i$ is a spin field of $i$-th copy of a $d = 2$ critical Ising model. In this case, the $\mathcal{S}^{\text{aux}}$ is the action of $m$-uncoupled Ising models.

All the above disorder distributions share the properties Eq. (2) but differ in their higher cumulants. A discussion on the effects of higher cumulants, based on numerical observations, can be found in [1,8].

As it is argued below, the consistency of our RG approach requires the $\sigma$ fields to have the following short distance expansion:

$$\sigma \times \sigma \to \text{Identity} + \sum_i C_{\sigma\sigma}^{\Phi_i} \, \Phi_i + \text{Irrelevant fields} \,, \qquad h_{\Phi_i} + 2h_\varepsilon > 2 \,. \tag{4}$$

In other words we assume that the only relevant fields $\Phi_i$ ($h_{\Phi_i} < 2$) that can appear in the $\sigma\sigma$ fusion rule, *i.e.* the corresponding structure constant is not vanishing $C_{\sigma\sigma}^{\Phi_i} \neq 0$, are the ones such that $\Phi_i \varepsilon^{(\alpha)} \varepsilon^{(\beta)}$ is irrelevant. The $\varepsilon^{(\alpha)}$ and $h_\varepsilon$ are respectively the $\alpha-$ Potts replica energy field and its conformal dimension, see below.

## 2.2 The effective model at the continuum limit

We initially consider the following action:

$$\mathcal{S} = \mathcal{S}^{\text{Potts}} + g_{LR}^0 \int d^2x \, \sigma(x)\varepsilon(x) \,, \tag{5}$$

where $\mathcal{S}^{\text{Potts}}$ is the conformal action describing the critical $q-$Potts model. Henceforth we indicate with $\langle \cdots \rangle$ the correlation functions of the critical Potts model. In a Lagrangian approach these correlation functions are calculated as functional averages with the weight exponential $\mathcal{S}^{\text{Potts}}$. We do not need to give here an explicit representation of such action. In the perturbation scheme used here, we just need to use the scaling dimension of its primary fields, known for a long time [22], and to compute certain four-point correlation functions. These latter can be computed using a Coulomb gas approach [15], see also Appendix (A). The field $\varepsilon(x)$ in Eq. (5) is the density energy which, by definition, is the scaling field coupled to the temperature and $\sigma(x)$ is the disorder field introduced above. The above action Eq. (5) describes then the continuum limit of a Potts model at temperature $J = J_c + \delta J(x)$, where $J_c$ is the critical temperature and $\delta J(x)$ are the random coupling fluctuations, $\delta J(x) \propto \sigma(x)$. The bare coupling $g_{LR}^0$ parametrizes the amplitude of these fluctuations, or in other terms, the strength of the disorder.

In the replica approach to quenched disorder, one is lead to consider the following action containing $n$-replicas of the original action [23] :

$$\mathcal{S}^{(n)} = \mathcal{S}^0 + \sum_{\alpha=1}^n g_{LR}^0 \int d^2x \, \sigma(x)\varepsilon^{(\alpha)}(x) \,,$$
$$\mathcal{S}^0 = \mathcal{S}^{\text{aux}} + \sum_{\alpha=1}^n \mathcal{S}^{(\alpha)\text{-Potts}} \,. \tag{6}$$

We carry out a perturbative expansion around the conformal action $\mathcal{S}^0$. The fact that the $\mathcal{S}^{\text{aux}}$ is still present in the above action is due to the fact that, as in [3], we do not integrate over the $\sigma$ variable, which remains therefore a dynamical variable. There are two main reasons for this. The first is that, until the moment we do not need to specify $\mathcal{S}^{\text{aux}}$, our results are valid for general disorder distributions, in particular non-Gaussian ones, with the only constraint to satisfy Eq. (4) and Eq. (2). The second motivation is that in doing so we have to deal with field theories with local interactions.

At the 1-loop order, the RG recursion relation are completely fixed by the OPE of the interaction $\sum_\alpha \sigma \varepsilon^\alpha$ term with itself [24]. This latter is straightforwardly determined by the Eq. (4) and the $\varepsilon \times \varepsilon$ OPE :

$$\varepsilon^\alpha \times \varepsilon^\alpha = \text{Identity} + \text{Irrelevant fields} \,. \tag{7}$$

The above relation comes from the fact that $\varepsilon$ has a vanishing null-state at second level [18], which in turn implies that there are only two primary fields produced by the fusion of $\varepsilon$ with

any other field. When fused with itself, the $\varepsilon$ field produces, besides the Identity field, another field which determines the corrections to the scaling of the thermal Potts observables. This latter field has scaling dimension greater than 2 for $q \leq 4$. Using Eq. (4) and Eq. (7), and in particular:

$$\sum_\alpha \sigma \varepsilon^{(\alpha)} \times \sum_\beta \sigma \varepsilon^{(\beta)} \rightarrow \cdots + C^{\Phi_i}_{\sigma,\sigma} \Phi_i \sum_{\alpha,\beta} \delta_{\alpha,\beta} + \cdots , \tag{8}$$

the terms $\Phi_i \varepsilon^\alpha \varepsilon^\beta$ being irrelevant by the condition in the Eq.(4), one has :

$$\sum_\alpha \sigma \varepsilon^\alpha \times \sum_\beta \sigma \varepsilon^\beta = n \times \text{Identity} + n \sum_i C^{\Phi_i}_{\sigma,\sigma} \Phi_i + \sum_{\substack{\alpha,\beta \\ \alpha \neq \beta}} \varepsilon^\alpha \varepsilon^\beta + \text{ Irrelevant fields} . \tag{9}$$

The Identity fields renormalizes the vacuum energy: it can be dismissed from the computation as it does not affect the correlation functions. Moreover, in the disorder limit $n \rightarrow 0$, the eventual relevant fields $\Phi_i$ appearing in the $\sigma\sigma$ fusion are not generated in the Eq.(9). They will not affect the RG equations describing the disordered system. On the other hand, the term $\sum_{\substack{\alpha,\beta \\ \alpha \neq \beta}} \varepsilon^\alpha \varepsilon^\beta$ which couples different replica have to be added to the action. We will refer to this term, which is produced by integrating over an uncorrelated disorder distribution [15], as the short-range term. The fact that the LR interaction generates, under a renormalization transformation, a short-range term is a very common feature of this problem, as discussed for instance in [3].

We are then lead consider the following action:

$$\mathcal{S}^{(n)} = \mathcal{S}^{\text{aux}} + \sum_{\alpha=1}^{n} \mathcal{S}^{(\alpha)\text{-Potts}} + \mathcal{S}^{\text{pert}} , \tag{10}$$

$$\mathcal{S}^{\text{pert}} = \sum_{\alpha=1}^{n} g^0_{LR} \int d^2x \, \sigma(x) \varepsilon^{(\alpha)}(x) + \sum_{\substack{\alpha,\beta=1 \\ \alpha \neq \beta}}^{n} g^0_{SR} \int d^2x \, \varepsilon^{(\alpha)}(x) \varepsilon^{(\beta)}(x) . \tag{11}$$

By dimensional analysis, one finds that the LR and SR perturbation term have respectively dimension $2 - \epsilon_{LR}$ and $2 - \epsilon_{SR}$ with:

$$\epsilon_{LR} = 2 - h_\sigma - h_\varepsilon , \qquad \epsilon_{SR} = 2 - 2h_\varepsilon . \tag{12}$$

In terms of the variables $a$ and $q$ one has :

$$\epsilon_{LR} = 1 - \frac{a}{2} + \frac{\epsilon_{SR}}{2} , \tag{13}$$

and :

$$\epsilon_{SR} = 4 - \frac{6\pi}{2\pi - \arccos\left(\frac{q-2}{2}\right)} = \frac{4}{3}\left(\frac{q-2}{\pi}\right) + O\left((q-2)^2\right) , \tag{14}$$

where we take the branch $\pi/2 \geq \arccos(x) \geq 0$ for $0 \leq x \leq 1$. We will consider the case where the perturbation are slightly relevant and perform a double expansion in $\epsilon_{LR}$ and $\epsilon_{LR}$. So, in our RG scheme we keep the spatial dimension fixed, $d = 2$, and we vary $q$ by using the regularization parameter $q - 2$. This is possible [15] as there is a family of $q$−Potts CFT points for general values of $q$, see also [25]. This is different from the RG regularization scheme of [13] which is based on the double expansion in $2 - a$ and in the space dimension $2 - d$.

We have to make assumptions on the relative magnitude of the two parameters $\epsilon_{SR}$ and $\epsilon_{LR}$ on which the double expansion is based. For $(q \neq 2)$-Potts model ($\epsilon_{SR} \neq 0$), we will in general consider the case where:

$$\epsilon_{SR}, \epsilon_{LR} \rightarrow 0 , \qquad \frac{\epsilon_{SR}}{\epsilon_{LR}} \rightarrow \text{ finite} . \tag{15}$$

We can equivalently carry out the RG protocol using one regularization parameter $\epsilon$, defined as:

$$\epsilon_{LR} = \epsilon, \qquad \epsilon_{SR} = s\,\epsilon, \tag{16}$$

with $s$ finite, and in particular $s = O(1)$. Notice that:

$$s = \frac{\epsilon_{SR}}{\epsilon_{LR}} = 2 - \frac{2-a}{\epsilon_{LR}} < 2, \quad \text{for} \quad a < 2, \ q \geq 2. \tag{17}$$

In our computation, the renormalizability of the theory is manifest in the fact that the 1-loop coefficients $\left(\text{of order } O\left(g_{SR}^2, g_{LR}^2, g_{SR}g_{LR}\right)\right)$ and the 2-loop coefficients $\left(\text{of order } O\left(g_{SR}^3, g_{LR}^3, g_{SR}^2 g_{LR}, g_{SR}g_{LR}^2\right)\right)$ of the RG recursion relations do not depend on $\epsilon$. These coefficients can eventually depend on the ratio $s$, as it will be our case, see Eq. (19) below.

We present in the following the main results of our RG computations. Our approach is an extension of the one introduced in [15, 26–28] for the short-range disorder. The details are given in the Appendices.

## 3 Renormalization group recursion relations

The main information on the RG flow, and therefore on the critical properties of the model, are encoded in the $\beta_{LR}$ and $\beta_{SR}$ functions, defined as:

$$\beta_{SR} = r\frac{d}{d\,r}\,g_{SR}, \qquad \beta_{LR} = r\frac{d}{d\,r}\,g_{LR}, \tag{18}$$

where $r$ is the RG scale parameter and $g_{SR}$ and $g_{LR}$ are the dimensionless renormalized couplings, see Eq. (A.45). Using the parametrization Eq. (16), we found in the $n \to 0$ limit, that:

$$\beta_{SR} = s\,\epsilon\,g_{SR} - 8\pi g_{SR}^2 + \pi\,g_{LR}^2 + 32\pi^2 g_{SR}^3 + \pi^2 \mathcal{B}(s) g_{LR}^2 g_{SR} + O\left(g^4\right), \tag{19}$$

$$\beta_{LR} = \epsilon\,g_{LR} - 4\pi g_{LR}g_{SR} + \pi^2 \mathcal{C}(s)\,g_{LR}^3 + 8\pi^2 g_{SR}^2 g_{LR} + O\left(g^4\right).$$

Above and henceforth, we use a short-cut notations $O(g^k)$ to indicate the $k-$order in a multi-variable expansion. For instance $O(g^3) = O(g_{SR}^3, g_{LR}^3, g_{SR}^2 g_{LR}, g_{SR}g_{LR}^2)$. The coefficient $\mathcal{B}$ appearing at order $O(g^3)$ is given by:

$$\mathcal{B}(s) = \frac{16\pi}{3\sqrt{3}}\frac{1-s}{(2-s)} - \frac{4\,s}{2-s}. \tag{20}$$

The other coefficient $\mathcal{C}$ appearing at order $O(g^3)$ is for the moment undetermined. Indeed this coefficient, defined in Eq. (A.44), depends on the fourth cumulant of the disorder, see Eq. (A.22) and Eq. (A.23). For its computation one has to specify more precisely the disorder distribution under consideration. We computed its value for a Gaussian disorder, see Eq. (D.6), and for particular instance of a non-Gaussian disorder distribution, see Appendix (D). In both cases, it is confirmed that $\mathcal{C}$ depends only on the ratio $s = \epsilon_{SR}/\epsilon_{LR}$.

In the following, we prefer to implicitly assume the validity of Eq. (15) and present the RG results in terms of $\epsilon_{SR}$ and $\epsilon_{LR}$ instead of using the parametrization Eq. (16).

### 3.1 Fixed points

We can now look for the main features of the RG flow. We start by the determination of its fixed points $(g_{SR}^*, g_{LR}^*)$ at which $\beta_{SR} = \beta_{LR} = 0$.

First, observe that $\beta_{LR} = g_{LR}(\cdots)$ and is expected to be proportional to $g_{LR}$ at all orders. This can be simply proven by observing that, in the perturbative expansion of the interaction

terms, only the ones containing an odd number of LR interactions contribute to the normalization of $g_{LR}$. If initially zero at the UV scale, the LR coupling remains zero, $g_{LR} = 0$, all along the RG flow. In this case our model reduces precisely to the one studied in [15]. There exists the (trivial) fixed point P with $g_{LR}^* = g_{SR}^* = 0$. The P point describes the critical pure Potts model and it is unstable in all directions, see below. Another fixed point is the SR critical point [15, 26] :

$$g_{SR}^{*,SR} = \frac{\epsilon_{SR}}{8\pi} + \frac{\epsilon_{SR}^2}{16\pi} + O(\epsilon^3), \qquad g_{LR}^{*,SR} = 0.$$ (21)

Let's now turn on the LR interaction, $g_{LR} \neq 0$. We find a new fixed point, henceforth referred to as the LR fixed point, which is located at

$$g_{SR}^{*,LR} = \frac{\epsilon_{LR}}{4\pi} + \frac{\epsilon_{LR}^2(\mathcal{C}+1)}{8\pi} - \frac{\epsilon_{LR}\epsilon_{SR}\,\mathcal{C}}{16\pi} + O(\epsilon^3),$$ (22)

$$g_{LR}^{*,LR} = \sqrt{2 - \frac{\epsilon_{SR}}{\epsilon_{LR}}} \left( \frac{\epsilon_{LR}}{2\pi} + \frac{-\mathcal{B}\epsilon_{LR}(2\epsilon_{LR} - \epsilon_{SR}) + \mathcal{C}\left(8\epsilon_{LR}^2 + \epsilon_{SR}^2 - 6\epsilon_{LR}\epsilon_{SR}\right) - 2\epsilon_{LR}\,\epsilon_{SR}}{16\pi\left(2 - \frac{\epsilon_{SR}}{\epsilon_{LR}}\right)} \right) + O(\epsilon^3).$$ (23)

It is important to observe that the LR fixed point remains at distance $O(\epsilon)$ from the origin provided the conditions Eq. (16) are satisfied. This justifies the perturbative approach to the computation of the corresponding critical exponents.

## 3.2 Stability of fixed points

The stability of a RG fixed point is established by linearizing the flow in the vicinity of that point, which leads to the $2 \times 2$ stability matrix:

$$M = \begin{pmatrix} \dfrac{\partial}{\partial\, g_{SR}} \beta_{SR}\left(g_{SR}^*, g_{LR}^*\right) & \dfrac{\partial}{\partial\, g_{LR}} \beta_{SR}\left(g_{SR}^*, g_{LR}^*\right) \\[2ex] \dfrac{\partial}{\partial\, g_{SR}} \beta_{LR}\left(g_{SR}^*, g_{LR}^*\right) & \dfrac{\partial}{\partial\, g_{LR}} \beta_{LR}\left(g_{SR}^*, g_{LR}^*\right) \end{pmatrix}.$$ (24)

The fixed point under consideration is stable if the real part of the eigenvalues is negative.

When $g_{LR}^* = 0$, the matrix $M$ is diagonal at all orders. This can be also be understood by recalling that $\beta_{LR} \propto g_{LR}$ and by further observing that $\partial/\partial_{g_{LR}} \beta_{SR} \propto g_{LR}$ as the terms that contributes to the normalization to $g_{SR}$ have an even number of LR interactions.

The P point, $g_{LR}^* = g_{SR}^* = 0$ has then the eigenvalues $\epsilon_{LR}$ and $\epsilon_{SR}$. It is unstable if either $\epsilon_{SR} > 0$ (or $2 < q \leq 4$) or if $\epsilon_{LR} > 0$ (or $a < 2$). Since we consider the case $2 \leq q \leq 4$ and $a < 2$ in the present work, the pure point is always unstable. In Fig. (1), the P point do not describe anymore the critical behavior of the system in this region of parameters.

For the SR fixed point Eq. (21), we find at the 2-loop expansion

$$\lambda_1^{SR} = -\epsilon_{SR} + \frac{\epsilon_{SR}^2}{2}, \qquad \lambda_2^{SR} = \epsilon_{LR} - \frac{\epsilon_{SR}}{2} - \frac{\epsilon_{SR}^2}{8}.$$ (25)

The first eigenvalue is always negative, $\lambda_1^{SR} < 0$, in the regime of parameters we are interested as $0 < \epsilon_{SR} \leq 1$ for $2 < q \leq 4$, see Eq. (14). The SR point is therefore stable provided that the second eigenvalue is negative, $\lambda_2^{SR} < 0$. This happens when:

$$a > 2 - \frac{\epsilon_{SR}^2}{4} = 2\left(1 - \frac{\epsilon_{SR}^2}{8}\right).$$ (26)

Let us comment on the above result. The SR correlation length exponent $\nu^{SR}$, computed in [15], and re-derived in Section (4.2), is:

$$\nu_{SR} = 1 + \frac{\epsilon_{SR}^2}{8} + O\left(\epsilon^3\right).\tag{27}$$

Using Eq. (27) in Eq. (26), we recover the extended Harris criterion [2] according to which the LR/SR change of stability occurs at $a = 2/\nu_{SR}$. This is also the equation defining the curve shown in Fig. (1). In RG terms, this criterion simply states that, at the SR point, the LR perturbation is irrelevant.

We consider now the LR point Eq. (22). Details of the analysis of the corresponding stability matrix are reported in Appendix (E).

We discuss first the Ising case, corresponding to $\epsilon_{SR} = 0$ (or $s = 0$ in the parametrization Eq. (16)). The two eigenvalues are:

$$\lambda_1^{LR,\text{Ising}} = -2\epsilon_{LR} + \frac{\epsilon_{LR}^2}{4}\left(8 + \mathcal{B}^{\text{Ising}} - 2\mathcal{C}\right) + 2i\epsilon_{LR}^{3/2},\tag{28}$$

$$\lambda_2^{LR,\text{Ising}} = -2\epsilon_{LR} + \frac{\epsilon_{LR}^2}{4}\left(8 + \mathcal{B}^{\text{Ising}} - 2\mathcal{C}\right) - 2i\epsilon_{LR}^{3/2},\tag{29}$$

where, from Eq. (20), one has:

$$\mathcal{B}(s=0) = \mathcal{B}^{\text{Ising}} = \frac{8\pi}{3\sqrt{3}}.\tag{30}$$

The eigenvalues have an imaginary term, in accordance with the findings of [13]. Some comments are in order. Concerning to the origin of the imaginary terms, we observe that, at the 1-loop order, the LR Ising stability matrix is not diagonalizable (note that $M$ is a real and not symmetric function) it has a single eigenvector, see Appendix. (E). At the 2-loop order, this is solved by creating a pair of complex eigenvalues, conjugate one to the other. A second observation is that the LR stability depends on the coefficient $\mathcal{C}$ and therefore on the fourth cumulant of the disorder distribution. For a Gaussian disorder, where Eq. (D.6) is verified, the real parts of the two eigenvalues, expressed in terms of $a$, take the form:

$$\text{Re}\left[\lambda_1^{LR,\text{Ising}}\right] = \text{Re}\left[\lambda_2^{LR,\text{Ising}}\right] = \frac{1}{2}(a-2)\,a.\tag{31}$$

This implies the existence of an interval, $a^* < a < 2$, with $a^* = 0$ for which the LR remain attractive. An analogous phenomena was observed in [13] where the interval of stability was $a^* < a < 2$ with $a^* \sim 1$. In [1], we show that below this value the system is attracted to an infinite disorder fixed point. The numerical simulations in [1] are consistent with a value of $a^* \sim 0.75$. The fact that our $a^* = 0$ is quite different from the numerical findings does not put in question the validity of our approach. Indeed $2 - a^* > 1$ and one cannot pretend to find a quantitative agreement with a small $2 - a$ expansion. However, the 2-loop computation shows a qualitative agreement with the fact that there should be a value $a^*$ below which the LR point loses stability. This is also confirmed for the Potts case, as we see below.

In the case of Potts, $\epsilon_{SR} \neq 0$, we obtain:

$$\lambda_1^{LR} = -2\epsilon_{LR} - \frac{\epsilon_{LR}^2}{2}\left(-2 + \mathcal{B} - 2\mathcal{C}\right) + \frac{\epsilon_{LR}^3}{\epsilon_{SR}}\left(4 + \mathcal{B} - 2\mathcal{C}\right),\tag{32}$$

$$\lambda_2^{LR} = -2\epsilon_{LR} + \epsilon_{SR} + \epsilon_{LR}^2\left(3 + \mathcal{B} - 2\mathcal{C}\right) - \frac{\epsilon_{LR}^3}{\epsilon_{SR}}\left(4 + \mathcal{B} - 2\mathcal{C}\right) - \frac{\epsilon_{SR}\epsilon_{LR}}{4}\left(\mathcal{B} - 2\mathcal{C}\right).\tag{33}$$

Again, in the Gaussian case Eq. (D.6) the formula get simplified. One finds that the LR point is attractive between:

$$a^* < a < \frac{2}{\nu_{SR}}, \qquad a^* = 2 - (2\epsilon_{SR})^{1/2} + \frac{5}{4}\epsilon_{SR} - \frac{1}{32\sqrt{2}}\epsilon_{SR}^{3/2}. \qquad (34)$$

For 3-Potts one has for instance $a^* = 1.6$. As in the Ising case, this prediction is qualitatively far from the numerical results in [1], where $a^* \sim 0.75$.

## 4 Critical exponents

We compute the effective central charge $c_{eff}^X$ and the correlation length exponent $\nu^X$ of the fixed points $X = \{SR, LR\}$. The values of $c_{eff}^{LR}$ and $\nu^{LR}$ are new results. To test the validity of our theory, the value of $c_{eff}^{LR}$ for the Ising model is compared to Monte Carlo observations.

Let us stress that we are assuming that the disordered fixed points have Virasoro symmetry. When the replicated action $\mathcal{S}^{(n)}$, see Eq. (10), has local conformal invariance, this assumption is well justified as the fixed point of the replicated theory is also conformal. In this case, one has to verify that the analytic continuation in the number of replica $n \to 0$ is safe, in particular that a replica symmetry breaking mechanism is not in place. This mechanism has been ruled out for the SR fixed point, see [29–31]. We are assuming here the replica symmetry is not broken for the LR fixed points too. When the replicated theory has not local conformal symmetry, such has in the case of Eq. (D.1), the assumption that the LR point has Virasoro symmetry appears optimistic. In [32,33] an analogous case has been considered, in which the ($d = 2$ and $d = 3$) critical Ising model is coupled to a Gaussian scale-invariant action. It was explicitly shown that the new RG fixed point does not enjoy local conformal symmetry.

Below we compare the theoretical predictions to the model studied in [1], where the $\mathcal{S}^{\text{aux}}$ action is the copy of $m-$Ising critical CFTs. We are therefore in the more safe case of a Virasoro symmetric replicated theory. Moreover, we provide here 1-loop order results, at which we expect the results not to be affected by the higher cumulants which distinguish between the different disorder distributions.

### 4.1 Central charge

The central charge $c$ is an important universal quantity that fixes the conformal algebra on which the theory is built [18]. The central charge determines the universal critical finite size corrections of many observables [34], the study of which allows to measure its value, see for instance [35]. Here we consider the finite size effects of the free energy $F$. Consider a critical statistical model defined on a torus of dimension $N \times L$. The central charge appears in the sub-leading and universal term of the large $L$ expansion of the free energy [36,37]

$$\frac{\beta_c F}{N L} = f_0 - \frac{c\pi}{6L^2} + O\left(L^{-4}\right), \qquad (35)$$

where $\beta_c$ is the critical temperature and $f_0$ is the free-energy density, which is a non-universal quantity.

For the pure model, the central charge $c^P$ is known exactly [21]. Its $\epsilon_{SR}$ expansion is:

$$c = c^P = \frac{1}{2} + \frac{7\epsilon_{SR}}{8} - \frac{9\epsilon_{SR}^2}{32} - \frac{9\epsilon_{SR}^3}{128} + O\left(\epsilon_{SR}^4\right). \qquad (36)$$

Recall that the $\epsilon_{SR} = 0$ corresponds to the Ising point where one recovers the well known result $c_{\text{Ising}}^P = 1/2$. In strict analogy with the pure case, the effective central charge $c_{eff}$ is

defined as the coefficient appearing in large $L-$expansion of the average free-energy:

$$\mathbb{E}\left[\frac{\beta_c F}{NL}\right] = f_0 - \frac{c_{eff}\,\pi}{6L^2} + O\left(L^{-4}\right). \tag{37}$$

In Appendix (B), we show how to compute the $c_{eff}^X$ for the $X = \{SR, LR\}$ fixed points. To compare with numerical results, the 1-loop order is sufficient: the numerical precision is not enough to probe 2-loop corrections.

At 1-loop order we obtain :

$$c_{eff}^{SR} = \frac{1}{2} + \frac{7\epsilon_{SR}}{8} - \frac{9\epsilon_{SR}^2}{32} - \frac{5\epsilon_{SR}^3}{128} + O\left(\epsilon_{SR}^4\right), \tag{38}$$

$$c_{eff}^{LR} = \frac{1}{2} + \frac{7\epsilon_{SR}}{8} - \frac{9\epsilon_{SR}^2}{32} - \frac{9\epsilon_{SR}^3}{128} - \frac{\epsilon_{LR}^3}{2} + \frac{3\,\epsilon_{LR}^2\epsilon_{SR}}{8} + O\left(\epsilon^4\right). \tag{39}$$

At the SR point, we recover the result of [27], while the effective charge at the LR point represents a new result. In particular, at the Ising point we have, expressed in terms of $a$:

$$c_{eff}^{LR,\text{Ising}} = \frac{1}{2} - \frac{1}{2}\left(1 - \frac{a}{2}\right)^3 + O\left(\epsilon_{LR}^4\right). \tag{40}$$

We show in Fig. (2) the comparison between our result Eq. (40) and the numerical results for the long-range Ising model. We refer the reader to [1] for a detailed definition of the lattice model used for simulations. In particular, the LR bond disordered Ising model is simulated by varying two parameters, the power decay exponent $a$ and the disorder strength $r$, see Eq. (2.10) of [1].[1] In Fig. (2), we show the effective central charge obtained from measurements of the averaged free energy on strips of size $N = 10^5$ and $L = \{4, \cdots, 8\}$ with periodic boundary conditions along the short direction. The measurements are done by averaging over $10^6$ strips. While for the uncorrelated bond disordered Potts model, the free-energy is a self-averaging observable [38], we have numerically checked that, for correlated disorder and for a fixed width $L$, the free energy per spin $f(L,N)$ follows a normal distribution as a function of $N$. Since the average over the disorder is obtained by taking a large $N$ limit (or equivalently by averaging over many $N \times L$ strips), the free energy is also self averaging over the disorder average. The $c_{eff}$, denoted $c(4,8)$ in the following, is obtained by fitting the results for different $L$ by the Eq. (37) while including a $L^{-4}$ correction. For the Ising model, a fit to this form gives $c(4,8) = 0.495998$ in place of the exact result $1/2$. This value is shown as a dashed line in Fig. (2). We also show the prediction Eq. (40) to which we subtract $0.5 - 0.495998 = 0.004002$. Next, we show our numerical results for the LR points. For each value of $a$, we compute $c(4,8)$ for the value of disorder strength $r$ with the less correction to the scaling in the magnetisation measurements as done in [1]. We employed the following values of disorder: $a/r = \{0.75/10, 1.0/10, 1.25/5, 1.50/3, 1.75/2, 2/2\}$. For $a \in \{1, 2\}$, the agreement between our measurement and the RG prediction Eq. (40) is excellent. For the smallest value of $a$ considered, $a = 0.75$, the effective central charge at the LR fixed point with $r = 10$ is the same as the one for the LR with an infinite disorder (also shown in Fig. (2) for all values of $a$). This is in agreement with our finding in [1] that $a^* \simeq 0.75$ is the lower limit of stable LR fixed point for the Ising model. Note that in the large $a$ limit, the LR with an infinite disorder corresponds to the percolation limit discussed in [39, 40].

We compared in this section the measured central charge with our prediction at the 1-loop order (39) for the Ising model with a nice agreement. The results for the $q > 2$ case, which contains an additional short range fixed point, will be presented in a subsequent work.

---

[1]This $r$ should not be confused with the renormalization group length scale.

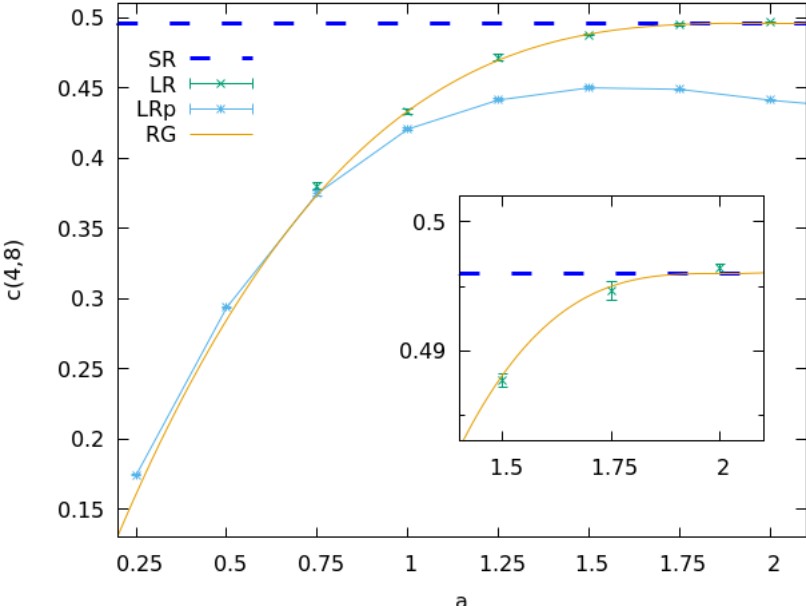

Figure 2: Effective central charge $c(4, 8)$, for the Ising model, measured at the LR fixed points indicated as LR in the key. This is compared with our prediction Eq. (40) shown as RG in the key. The inset shows a magnified view for $a$ close to 2. We also show the effective central charge for the LR model with infinite disorder, indicated by LRp in the key. SR, in the key, is instead the value measured for the Ising model at the SR fixed point.

## 4.2 Correlation length exponent

In the replica approach, the average of the two-point function $\mathbb{E}\left[\langle\varepsilon(x)\varepsilon(0)\rangle\right]$ is expressed as:

$$\mathbb{E}\left[\langle\varepsilon(x)\varepsilon(0)\rangle\right] \propto \lim_{n\to 0}\frac{1}{n}\left\langle\!\!\left\langle\sum_{\alpha=1}^{n}\varepsilon^{(\alpha)}(x)\sum_{\beta=1}^{n}\varepsilon^{(\beta)}(0)\right\rangle\!\!\right\rangle, \tag{41}$$

where $\langle\!\langle\cdots\rangle\!\rangle$ is the average over the whole replicated action Eq. (10). Accordingly, to compute the power law decay exponent of $\mathbb{E}\left[\langle\varepsilon(x)\varepsilon(0)\rangle\right]$, i.e. the energy scaling exponent at one of the RG fixed points, we study the renormalization of the replica symmetric field

$$\varepsilon^{(\mathrm{sym})} = \sum_{\alpha=1}^{n}(\varepsilon^{\alpha}). \tag{42}$$

It follows directly from the OPE between $\varepsilon^{(\mathrm{sym})}$ and the SR and LR interaction terms, that the RG flow of the $n$-replicated action produces a mixing between the $\varepsilon$ and $\sigma$ fields. In the RG procedure this means that the $2\times 2$ matrix $Z_{\varepsilon^{(sym)},\sigma}$:

$$\begin{pmatrix}\left(\varepsilon^{(\mathrm{sym})}\right)'\\\sigma'\end{pmatrix} = Z_{\varepsilon^{(sym)},\sigma}\begin{pmatrix}\varepsilon^{(\mathrm{sym})}\\\sigma\end{pmatrix}, \tag{43}$$

where the $\left(\varepsilon^{(\mathrm{sym})}\right)'$ and $\sigma'$ are the renormalized fields, has to be taken into account. From the analysis of the combinatorial factors associated to the RG expansion diagrams, we observed that the matrix $Z_{\varepsilon^{(sym)},\sigma}$ takes the form:

$$Z_{\varepsilon^{(sym)},\sigma} = \begin{pmatrix}Z_{\varepsilon} & Z_{12}\\n\times Z_{21} & n\times Z_{22}\end{pmatrix}, \tag{44}$$

where $Z_\varepsilon, Z_{12}, Z_{21}, Z_{22}$ are non-vanishing contributions. Therefore, in the limit $n \to 0$ the $\varepsilon'$ does not get mixed with $\sigma$. The renormalization of $\varepsilon$ depends only on $Z_\varepsilon$.

We computed $Z_\varepsilon$, at 2-loop order that, in the $n \to 0$ limit, is:

$$Z_\varepsilon = 1 - 4\pi\, g_{SR}^0\, \frac{r^{\epsilon_{SR}}}{\epsilon_{SR}} + 4\pi^2 \left(g_{SR}^0\right)^2 \frac{r^{2\epsilon_{SR}}}{\epsilon_{SR}} \left[1 + \frac{6}{\epsilon_{SR}}\right] + 2\pi^2 \left(g_{LR}^0\right)^2 \frac{r^{2\epsilon_{LR}}}{2\epsilon_{LR}} \left[\frac{-2}{2\epsilon_{LR} - \epsilon_{SR}} + \frac{\mathcal{B}}{4}\right],$$
(45)

where $\mathcal{B}$ is given in Eq. (20). The function

$$\gamma_\varepsilon = r \frac{d}{dr} \log Z_\varepsilon \,,$$
(46)

entering in the Callan-Symanzik equations [15], is written in terms of the normalized variables, as

$$\gamma_\varepsilon = -4\pi g_{SR} + 8\pi^2 g_{SR}^2 + \pi^2 g_{LR}^2 \frac{\mathcal{B}}{2} \,.$$
(47)

The renormalized energy scaling exponent $h_\varepsilon'^{,X}$ at the fixed point $X = \{SR, LR\}$ is given by:

$$h_\varepsilon'^{,X} = h_\varepsilon - \gamma_\varepsilon \left(g_{SR}^{*,X}, g_{LR}^{*,X}\right) \,.$$
(48)

This also gives the thermal exponent at the $X$ point:

$$\frac{1}{\nu^X} = 2 - h_\varepsilon'^{,X} \,.$$
(49)

Using the expansion:

$$h_\varepsilon = 1 - \frac{\epsilon_{SR}}{2} + O\left(\epsilon_{SR}^3\right) \,,$$
(50)

and the location Eq. (21) of the SR fixed point in Eq. (48), we recover the result of [15]:

$$h_\varepsilon'^{,SR} = 1 + \frac{\epsilon_{SR}^2}{8} \,,$$
(51)

which by the Eq. (49), gives the result Eq. (27).

The new result is the LR thermal exponent. Using the LR fixed point location Eq. (22) in Eq. (47), we obtain:

$$h_\varepsilon'^{,LR} = h_\varepsilon + \epsilon_{LR} + \frac{1}{4}\left(\mathcal{C} - \frac{\mathcal{B}}{2}\right)\epsilon_{LR}(2\epsilon_{LR} - \epsilon_{SR}) + O\left(\epsilon^3\right) \,.$$
(52)

The Ising model is obtained by setting $\epsilon_{SR} = 0$ in the above equation. In terms of the disorder decay parameter $a$, we have:

$$\text{Ising:} \quad h_\varepsilon'^{,LR} = 2 - \frac{a}{2} + \frac{1}{2}\left(\mathcal{C} - \frac{\mathcal{B}}{2}\right)\left(1 - \frac{a}{2}\right)^2 + O\left((2-a)^3\right) \,.$$
(53)

We can see that, in case of a Gaussian disorder where $2\mathcal{C} = \mathcal{B}$, see Eq. (D.6), we find, by using Eqs. (12) and (13), that $\nu^{LR} = 2/a$. Otherwise, our result shows a violation of the Weinrib-Halperin conjecture at second loop order if $2\mathcal{C} \neq \mathcal{B}$. We have considered in Appendix (D.2) a non-Gaussian disorder where the $\mathcal{C}$ can be computed. The corresponding result Eq. (D.10) support the fact that the relation $\nu^{LR} = 2/a$ is violated at 2-loop order, see Eq. (D.11).

Some comments are in order. The Weinrib-Halperin conjecture is equivalent to the condition $h_\sigma + h_\varepsilon'^{,LR} = 2$. A similar relation, denoted as the shadow relation, was proven in [32,33]. As mentioned previously, these works considered a critical Ising model coupled via its spin field to a Gaussian field. The corresponding RG recursion relation describes a one-coupling

flow where the lowest non-vanishing contribution appears at the second loop order. Although different, this model and the one studied here with the distribution (D.1) show some analogies. The shadow relation is ultimately related to the non-locality of the Gaussian action, on the basis of which one can show that the Gaussian field does not renormalize [32, 33]. This is also the property used in [3] to prove that the Weinrib-Halperin conjecture is valid at all perturbation orders. Let us finally comment on the argument given in [4] which, in the form it was stated, seemed to be valid for general disorder distribution. This is why most of the subsequent numerical works compared their measured exponent $\nu^{LR}$ to the Weinrib-Halperin prediction without paying too much attention on the fact they were actually considering non-Gaussian disorder. The Weinrib argument can be reformulated by imposing the irrelevance of the field $\sigma \varepsilon^{LR}$. An important observation is that the term $\sigma \varepsilon^{LR}$ is not anymore a scaling field at the LR point, in other words is not an eigenvector of the RG transformation. This provides a loophole in the Weinrib argument [4].

## 5 Conclusion

We studied by RG methods the two-dimensional $q-$Potts model with long-range quenched bond disorder. We focused on the region where $q \geq 2$ and where the decaying exponent $a$ is smaller than the space dimension, $a < 2$. This region is characterized by the fact that both the short-range and the long-range properties of the disorder distribution are relevant. Using a replica approach, we carried out a 2-loop order RG calculation based on a perturbed conformal field theory Eq. (10). Our RG procedure is based on a double expansion in the small parameters $2-a$ and $q-2$, with $(2-a)/(q-2)$ finite.

We compute the renormalization of the couplings $g_{SR}$ and $g_{LR}$ appearing in the replicated theory Eq. (6). These results are encoded in the recursion relation Eq. (19). We showed that the RG flow has two non-trivial fixed points, the SR and LR points, one dominated by the short-range interactions and the other by the long-range ones. We showed that the stability of these points is perfectly consistent with the phase diagrams studied in [1] and with the findings of previous works, see in particular [13] for the $q = 2$ Ising point. In particular, for any $q \geq 2$, our RG calculation predicts an interval $a^*(q) < a < 2/\nu^{SR}(q)$ where the LR point is stable.

We have provided here, for the first time, exact results on the effects of higher cumulants of the disorder distributions. The main result is the computation of the $\nu^{LR}$ thermal exponent for the Potts, Eq. (52), and for the Ising model, Eq. (53). We showed that the Weinrib-Halperin conjecture $\nu^{LR} = 2/a$ is valid for a Gaussian disorder but can be violated for other disorder distributions differing in the fourth cumulant. To show this, we provided an example of a non-Gaussian disorder distribution, see Appendix. (D), for which $\nu^{LR}$ is given in Eq. (D.11).

To test the validity of our theory, we computed the LR effective central charge Eq. (39). In Fig. (2) we showed that the theoretical predictions match very well with the transfer-matrix calculations for the long-range Ising model.

## Acknowledgments

We are grateful to Pierre Le Doussal for very crucial suggestions. We thank Kay Wiese and Emilio Trevisani for useful discussions.

# A  Couplings renormalization

We give here the main details of our RG computation. For an exhaustive explication of the RG scheme used here, we refer the reader to Chapter 5 of [24] for the basic ideas and to [15, 29, 41, 42] for the main technicalities behind this approach.

A RG protocol based on perturbation theory is composed by the following main steps:

- We expand the perturbation term:

$$
e^{S^{\text{pert}}} = 1 + \sum_{\alpha=1}^{n} g_{LR}^0 \int d^2x \, \sigma(x)\varepsilon^{(\alpha)}(x) + \sum_{\alpha\neq\beta=1}^{n} g_{SR}^0 \int d^2x \, \varepsilon^{(\alpha)}(x)\varepsilon^{(\beta)}(x)
$$
$$
+ \frac{(g_{LR}^0)^2}{2} \sum_{\alpha,\beta=1}^{n} \int d^2x \, \sigma(x)\varepsilon^{(\alpha)}(x) \int d^2y \, \sigma(y)\varepsilon^{(\beta)}(y) + \cdots \quad\quad \text{(A.1)}
$$

- Order by order, we integrate the fluctuations over the distances $|x|$, $1 < |x| < r$. We assume here that 1 is the initial UV cutoff and $r$ is the RG scale parameter. The UV cutoff will be always omitted in the formulas as it does not affect the RG recursion relations.

- We scale the system back to the initial UV cutoff.

- We compute the renormalized couplings $g_{LR}(r)$, $g_{SR}(r)$ in terms of the bare ones $g_{LR}^{(0)}(r)$, $g_{SR}^{(0)}(r)$.

We carry out the above procedure by using a real space RG. This is particularly adapted when one deals with perturbations around a conformal (in general not a free-field) action. This can be compared to previous RG approaches to long-range disordered models, where the unperturbed action was a free-field action: for instance a vector [2] or a tensor of free-scalar-fields [4], or again a Majorana free fermion [13]. The RG transformations in these previous works were carried out in the momentum space by using the form of the free-theory bare propagator. Here we use instead the bootstrap data of the unperturbed CFT, that provides the space dependence of the three and the four-point functions of primary fields.

It is a very general result in perturbed CFTs [24] that the 0−loop order and 1−loop order RG recursion relation are determined by the scaling dimensions of the relevant perturbation fields and by theirs OPE. In the following, we re-derive this result by specifying it to our case. This will allow also to extend the method to reach the 2−loop order.

## A.1  0-loop order

Let's consider for instance the following term in the expansion:

$$
g_{LR}^0 \int_{(1)} d^2x \, \sigma(x)\varepsilon(x), \quad\quad \text{(A.2)}
$$

where $\int_{(1)}$ means that one integrates over $|x| > 1$. Integrating between 1 and $r$, the fastest modes of the field $\sigma(x)\varepsilon(x)$ give vanishing contributions, thus leaving:

$$
g_{LR}^0 \int_{(r)} d^2x \, \sigma(x)\varepsilon(x). \quad\quad \text{(A.3)}
$$

We have therefore obtained the same term as before but with a rescaled UV cutoff. Now we have to rescale back to the initial cutoff using the transformations:

$$
x \rightarrow x' = \frac{x}{r}, \quad\quad \sigma(x)\varepsilon(x) \rightarrow r^{-h_{LR}}\sigma(x')\varepsilon(x'), \quad\quad \text{(A.4)}
$$

where we used the fact that $\sigma\varepsilon$ is a scaling field with dimension $h_{LR} = h_\sigma + h_\varepsilon$. The term Eq. (A.3) transforms to:

$$g_{LR}^0 r^{\epsilon_{LR}} \int_{(1)} d^2x\, \sigma(x)\varepsilon(x). \tag{A.5}$$

We have therefore that the coupling gets normalized as:

$$g_{LR}^0 \to g_{LR} = g_{LR}^0\, r^{\epsilon_{LR}}. \tag{A.6}$$

Notice that the renormalized coupling $g_{LR}$ is dimensionless.

## A.2  1-loop order

Starting from the first order, the contributions to the RG renormalization come from the interactions between perturbative fields when these approach each other at distances smaller than the RG scale, $r$. In particular, the first order contributions originate from the contribution of two colliding perturbative fields.

The contribution to $g_{SR}$ already considered in [15] is given by the term appearing in the first order expansion:

$$\frac{g_{SR}^2}{2} \sum_{\substack{\alpha,\beta=1 \\ \alpha\neq\beta}}^n \sum_{\substack{\rho,\eta=1 \\ \rho\neq\eta}}^n \int d^2x \int_{|y-x|<r} d^2y\, \varepsilon^{(\alpha)}(x)\varepsilon^{(\beta)}(x)\, \varepsilon^{(\rho)}(y)\varepsilon^{(\eta)}(y). \tag{A.7}$$

The contributions to $g_{SR}$ is given by the terms in the above sum with one pair of energy fields in the same replica, for instance $\beta = \rho$. There are $4(n-2)$ of such terms, each contribution with an integral of type:

$$\int d^2x\, \varepsilon^{(\alpha)}(x)\, \varepsilon^{(\rho)}(x) \int_{|y-x|<r} d^2y\, \langle\varepsilon(x)\varepsilon(y)\rangle = 2\pi \int d^2x\, \varepsilon^{(\alpha)}(x)\, \varepsilon^{(\rho)}(x) \int^r dy\, y^{1-2h_\varepsilon}$$

$$= \pi\frac{r^{2\epsilon_{SR}}}{\epsilon_{SR}} \int d^2x\, \varepsilon^{(\alpha)}(x)\, \varepsilon^{(\beta)}(x), \tag{A.8}$$

where, in the last line, we used the transformation $\varepsilon\varepsilon \to r^{-2h_\varepsilon}\varepsilon\varepsilon$ to set back the old UV scale. So, we have the contribution $\mathcal{I}_{g_{SR}^2}$ to $g_{SR}$ coming from Eq. (A.7):

$$\mathcal{I}_{g_{SR}^2} = 4\pi(n-2)\left(g_{SR}^{(0)}\right)^2 \frac{r^{2\epsilon_{SR}}}{\epsilon_{SR}}. \tag{A.9}$$

With respect of [15], at first order we have to consider two additional diagrams. One is a contribution to the normalization of $g_{SR}$ and it is:

$$\frac{\left(g_{LR}^0\right)^2}{2} \sum_{\alpha,\beta=1}^n \int d^2x \int_{|y-x|<r} d^2y\, \sigma(x)\varepsilon^{(\alpha)}(x)\, \sigma(y)\varepsilon^{(\beta)}(y). \tag{A.10}$$

When $\alpha = \beta$, the above terms renormalize to the identity and we discard them (they normalize the identity operator, *i.e.* a constant that multiplies the partition function). Let's fix $\alpha \neq \beta$. In that case we have:

$$\frac{\left(g_{LR}^0\right)^2}{2} \int d^2x\, \varepsilon^{(\alpha)}(x)\, \varepsilon^{(\beta)}(x) \int_{|y-x|<r} d^2y\, \mathbb{E}[\sigma(x)\sigma(y)]$$

$$= 2\pi\frac{\left(g_{LR}^0\right)^2}{2} \int d^2x\, \varepsilon^{(\alpha)}(x)\, \varepsilon^{(\beta)}(x) \int^r dy\, y^{1-2h_\sigma}$$

$$= \pi\left(g_{LR}^0\right)^2 \frac{r^{2\epsilon_{LR}}}{2\epsilon_{LR}-\epsilon_{SR}} \int d^2x\, \varepsilon^{(\alpha)}(x)\, \varepsilon^{(\beta)}(x), \tag{A.11}$$

where again, in the last line, we used the transformation $\varepsilon\varepsilon \to r^{-2h_\varepsilon}\varepsilon\varepsilon$ to set back the old UV scale. We have a contribution $\mathcal{I}_{g_{LR}^2}$ to $g_{SR}$ coming from the term of Eq. (A.10):

$$\mathcal{I}_{g_{LR}^2} = \pi\left(g_{LR}^0\right)^2 \frac{r^{2\epsilon_{LR}}}{2\epsilon_{LR} - \epsilon_{SR}}. \tag{A.12}$$

The second diagram is of the form:

$$g_{LR}^0 g_{SR}^0 \sum_{\substack{\alpha,\beta,\eta=1 \\ \beta\neq\eta}}^{n} \int d^2x \int_{|y-x|<r} d^2y\, \sigma(x)\varepsilon^{(\alpha)}(x)\, \varepsilon^{(\beta)}(y)\varepsilon^{(\eta)}(y). \tag{A.13}$$

Let's fix a $\alpha = \beta$, one gets:

$$g_{LR}^0 g_{SR}^0 \int d^2x\, \sigma(x)\varepsilon^{(\eta)}(x)\int_{|y-x|<r} dy\, \left\langle \varepsilon^{(\alpha)}\varepsilon^{(\alpha)} \right\rangle$$

$$= 2\pi g_{LR}^0 g_{SR}^0 \int d^2x\, \sigma(x)\varepsilon^{(\eta)}(x)\int_{|y-x|<r} dy\, |y|^{1-2h_{SR}}$$

$$= 2\pi g_{LR}^0 g_{SR}^0 \frac{r^{\epsilon_{SR}+\epsilon_{LR}}}{\epsilon_{SR}} \int d^2x\, \sigma(x)\varepsilon^{(\eta)}(x), \tag{A.14}$$

where, again, in the last line we used the transformation of the primary $\sigma(x)\varepsilon^{(\eta)}$ to set back the UV cutoff. There are $2(n-1)$ of such diagrams, so the expansion term Eq. (A.13) contributes with:

$$\mathcal{I}_{g_{SR}g_{LR}} = 4\pi(n-1)\frac{r^{\epsilon_{LR}+\epsilon_{SR}}}{\epsilon_{SR}}, \tag{A.15}$$

to the renormalization of $g_{LR}$.

So, collecting the 0 and 1−loop results, we have:

$$g_{SR} = r^{\epsilon_{SR}} g_{SR}^{(0)} + \mathcal{I}_{g_{SR}^2} + \mathcal{I}_{g_{LR}^2}, \tag{A.16}$$

$$g_{LR} = r^{\epsilon_{LR}} g_{LR}^{(0)} + \mathcal{I}_{g_{SR}g_{LR}}. \tag{A.17}$$

In the $n \to 0$, limit this becomes:

$$g_{SR} = r^{\epsilon_{SR}}\left( g_{SR}^{(0)} - 8\pi\left(g_{SR}^{(0)}\right)^2 \frac{r^{\epsilon_{SR}}}{\epsilon_{SR}} + \pi\left(g_{LR}^{(0)}\right)^2 \frac{r^{2\epsilon_{LR}-\epsilon_{SR}}}{2\epsilon_{LR}-\epsilon_{SR}} + O\left(g^3\right)\right), \tag{A.18}$$

$$g_{LR} = r^{\epsilon_{LR}}\left( g_{LR}^{(0)} - 4\pi\, g_{SR}^{(0)} g_{LR}^{(0)} \frac{r^{\epsilon_{LR}}}{\epsilon_{LR}} + O\left(g^3\right)\right). \tag{A.19}$$

Using the above equations in Eq. (18), one finds the results of Eq. (19).

## A.3 2-loop order

The 2-loop order is very important to draw conclusions from an approximated RG computations. The results at this order depend on how three fields interact at low distances and this probes the four-point correlation functions of the unperturbed CFT. In CFT, the four-point correlation function encodes the spectrum of the theory as well as its structure constant. This is also the reason why it is the central object in the bootstrap approach. There are many examples in which, for instance, properties that are valid at first loop order are broken at the second loop one. An important example is given here and it concerns $\nu^{LR}$, see Section (4.2).

The diagrams appearing at the 2-loop order can be reduced to these four integrals:

$$\mathcal{I}_1 = \int_{|y|<r} d^2 y \int_{|z|<r} d^2 z \, \mathbb{E}[\sigma(0)\sigma(y)]\langle\varepsilon(y)\varepsilon(z)\rangle, \tag{A.20}$$

$$\mathcal{I}_2 = \int_{|y|<r} d^2 y \int_{|z|<r} d^2 z \, \langle\varepsilon(0)\varepsilon(y)\varepsilon(z)\varepsilon(\infty)\rangle \, \mathbb{E}[\sigma(y)\sigma(z)], \tag{A.21}$$

$$\mathcal{J}_1 = \int_{|y|<r} d^2 y \int_{|z|<r} d^2 z \, \langle\varepsilon(y)\varepsilon(z)\rangle \, \mathbb{E}[\sigma(0)\sigma(y)\sigma(z)\sigma(\infty)], \tag{A.22}$$

$$\mathcal{J}_2 = \int_{|y|<r} d^2 y \int_{|z|<r} d^2 z \, \langle\varepsilon(0)\varepsilon(y)\varepsilon(z)\varepsilon(\infty)\rangle \, \mathbb{E}[\sigma(0)\sigma(y)\sigma(z)\sigma(\infty)]. \tag{A.23}$$

One has to expand these integrals in series of $\epsilon_{LR}$ and $\epsilon_{SR}$. For a general disorder distribution, we can give the result of $\mathcal{I}_1$ and $\mathcal{I}_2$ that depend only on the second cumulant of the disorder distribution while for the last two integrals, one has to specify also the fourth cumulant.

For the first integral $\mathcal{I}_1$, the leading term in the $\epsilon_{LR}, \epsilon_{SR}$ expansion is given by:

$$\mathcal{I}_1 \to 2\pi \int_{|z|<r} d\,z|z|^{-1+2\epsilon_{LR}} \int d^2 y |y|^{-2h_\sigma}|1-y|^{-2h_\varepsilon}. \tag{A.24}$$

Using the formula (see Eq. (7.4.10) in [43]):

$$\int d^2 y |y|^{2a}|1-y|^{2b} = \pi \frac{\Gamma[1+a]\Gamma[1+b]\Gamma[-1-a-b]}{\Gamma[-a]\Gamma[-b]\Gamma[2+a+b]}, \tag{A.25}$$

one extracts the leading term in the $\epsilon_{LR}, \epsilon_{SR}$ expansion:

$$\mathcal{I}_1(\epsilon_{SR}, \epsilon_{LR}) = \pi^2 \left[\frac{8\,\epsilon_{LR}}{\epsilon_{SR}(2\epsilon_{LR}-\epsilon_{SR})}\right] \frac{r^{2\epsilon_{LR}}}{2\epsilon_{LR}}. \tag{A.26}$$

As far the integral $\mathcal{I}_2$ is concerned, we use the Coulomb Gas representation of the Potts correlation functions [15, 44–46]:

$$\langle\varepsilon(0)\varepsilon(1)\varepsilon(y)\varepsilon(\infty)\rangle = -\frac{2\Gamma\left[-\frac{2}{3}\right]^2}{\sqrt{3}\Gamma\left[-\frac{1}{3}\right]^4}|y|^{\epsilon_{SR}-2}|y-1|^{\frac{4}{3}-\frac{1}{3}\epsilon_{SR}}$$
$$\times \int d^2 u \, |u|^{4-2\epsilon_{SR}}|u-1|^{-\frac{8}{3}+\frac{2}{3}\epsilon_{SR}}|u-y|^{-\frac{8}{3}+\frac{2}{3}\epsilon_{SR}}. \tag{A.27}$$

One has to evaluate the multi-complex integral:

$$\mathcal{I}_2 \to 2\pi \int_{|z|<r} d\,|z||z|^{-1+2\epsilon_{LR}} \int d^2 y |y|^{-2+\epsilon_{SR}}|1-y|^{-\frac{2}{3}-\frac{4}{3}\epsilon_{SR}+2\epsilon_{LR}}$$
$$\times \int d^2 u \, |u|^{4-2\epsilon_{SR}}|u-1|^{-\frac{8}{3}+\frac{2}{3}\epsilon_{SR}}|u-z|^{-\frac{8}{3}+\frac{2}{3}\epsilon_{SR}}. \tag{A.28}$$

The above integral can be computed using the techniques explained in Appendix (D) of [15]. The only difference with the integrals computed there is the exponent of the $|y-1|$ term. We obtain the result:

$$\mathcal{I}_2(\epsilon_{SR}, \epsilon_{LR}) = \pi^2 \left[\frac{8}{\epsilon_{SR}} + \mathcal{B}(s)\right] \frac{r^{2\epsilon_{LR}}}{2\epsilon_{LR}}, \tag{A.29}$$

where we recall that $s = \epsilon_{SR}/\epsilon_{LR}$ and $\mathcal{B}(s)$ is defined in Eq. (20).

Let's first re-derive the contribution to $g_{SR}$ coming only by the SR terms, which has been already computed in [15]. At the third order of the perturbative expansion one has the $(g_{SR}^{(0)})^3$ term:

$$\frac{\left(g_{SR}^{(0)}\right)^3}{6} \sum_{\substack{\alpha,\beta=1 \\ \alpha\neq\beta}}^n \sum_{\substack{\gamma,\iota=1 \\ \gamma\neq\iota}}^n \sum_{\substack{\eta,\rho=1 \\ \eta\neq\rho}}^n \int d^2x \int_{\substack{|y-x|<r \\ |z-x|<r}} d^2y \, d^2z \, \varepsilon^\alpha(x)\varepsilon^{(\beta)}(x) \, \varepsilon^{(\gamma)}(y)\varepsilon^{(\iota)}(y) \, \varepsilon^{(\eta)}(z)\varepsilon^{(\rho)}(z). \quad \text{(A.30)}$$

In the above sum one has to distinguish the terms where there are two pairs of energy fields in the same replica, for instance when $\alpha = \gamma$ and $\iota = \eta$ with $\beta \neq \rho,\iota$. There are $3 \times 8 \times (n-2)(n-3)$ of such terms, each of them contributing via an the integral Eq. (A.20) where $\mathcal{E}[\sigma(0)\sigma(y)]$ is replaced by $\langle\varepsilon(0)\varepsilon(y)\rangle$. Using the result Eq. (A.26) with $\epsilon_{LR} \to \epsilon_{SR}$, one has the contribution:

$$\left(g_{SR}^{(0)}\right)^3 4(n-2)(n-3)\mathcal{I}_1(\epsilon_{SR},\epsilon_{SR}) \to \left(g_{SR}^{(0)}\right)^3 16\,\pi^2\,(n-2)(n-3)\frac{r^{2\epsilon_{SR}}}{\epsilon_{SR}^2}. \quad \text{(A.31)}$$

Another contribution comes from the terms where one has a pair and a triple of fields with the same replica, for instance when $\alpha = \gamma = \eta$ and $\iota = \rho \neq \beta$. There are $3 \times 8 \times (n-2)$ of such terms, each of this contributing with an integral Eq. (A.21) where, again, $\mathbb{E}[\sigma(0)\sigma(y)]$ is replaced by $\langle\varepsilon(0)\varepsilon(y)\rangle$. Using Eq. (A.29) with $\epsilon_{LR} \to \epsilon_{SR}$ and $\mathcal{B}(s=1) = -4$, see Eq. (20), one has:

$$\left(g_{SR}^{(0)}\right)^3 4(n-2)\mathcal{I}_2(\epsilon_{SR},\epsilon_{SR}) \to \left(g_{SR}^{(0)}\right)^3 4\,\pi^2\,(n-2)\left[\frac{8}{\epsilon_{SR}} - 4\right]\frac{r^{2\epsilon_{SR}}}{2\epsilon_{SR}}. \quad \text{(A.32)}$$

The term with two triple of energy fields with equal replica indexes, $\alpha = \gamma = \eta$ and $\beta = \iota = \rho$, is associated to an integral which is of order $O(1)$, [15]. These diagrams do not contribute to the $\beta$ function. The contribution $\mathcal{I}_{g_{SR}^3}$ of Eq. (A.30) is then:

$$\mathcal{I}_{g_{SR}^3} = 4\left(g_{SR}^{(0)}\right)^3 \left[(n-2)(n-3)\mathcal{I}_1(\epsilon_{SR},\epsilon_{SR}) + (n-2)\mathcal{I}_2(\epsilon_{SR},\epsilon_{SR})\right]. \quad \text{(A.33)}$$

We will present now the new contributions, generated by the LR coupling, to the renormalization of $g_{SR}$. At the third order of expansion, one finds the term:

$$\frac{g_{SR}^{(0)}\left(g_{LR}^{(0)}\right)^2}{2} \sum_{\alpha=1}^n \sum_{\beta=1}^n \sum_{\substack{\eta,\rho=1 \\ \eta\neq\rho}}^n \int d^2x \int_{|y-x|<r} d^2y \int_{|z-x|<r} d^2z \, \sigma(x)\varepsilon^{(\alpha)}(x)\sigma(y)\varepsilon^{(\beta)}(y) \, \varepsilon^{(\eta)}(z)\varepsilon^{(\rho)}(z).$$

$$\text{(A.34)}$$

Among all the terms in the triple sum, we can identity two families of terms which renormalize $g_{SR}$. The first type corresponds to the terms where only two energy fields, at different points, belong to the same Potts replica. Because of the small distance expansions,

$$\sigma(x)\sigma(y) \to \mathbb{E}[\sigma(x)\sigma(y)] \times \text{Identity}, \qquad \varepsilon(y)\varepsilon(z) \to \langle\varepsilon(x)\varepsilon(y)\rangle \times \text{Identity}, \quad \text{(A.35)}$$

these terms renormalizes $g_{SR}$ with a contribution given by the integral $\mathcal{I}_1$ in Eq. (A.20). By combinatorial analysis, one finds that all these cases produces $4(n-2)$ terms $\sum_{\substack{\alpha,\eta=1 \\ \alpha\neq\eta}}^n \varepsilon^{(\alpha)}\varepsilon^{(\eta)}$.

The other type corresponds to the situation in which three energy fields at three different points are in the same Potts replica. Their contribution is expressed via the integral $\mathcal{I}_2$ as can be seen from the small distance behavior [15]:

$$\sigma(x)\sigma(y) \to \mathbb{E}[\sigma(x)\sigma(y)] \times \text{Identity}, \quad \varepsilon(x)\varepsilon(y)\varepsilon(z) \to \langle\varepsilon(x)\varepsilon(y)\varepsilon(z)\varepsilon(\infty)\rangle\,\varepsilon(x). \quad \text{(A.36)}$$

They generate $2 \times \sum_{\substack{\alpha,\eta=1 \\ \alpha \neq \eta}}^{n} \varepsilon^{(\alpha)} \varepsilon^{(\eta)}$ interaction terms. So we have the contribution $\mathcal{I}_{g_{SR} g_{LR}^2}$ of Eq. (A.34) to $g_{SR}$:

$$\mathcal{I}_{g_{SR} g_{LR}^2} = g_{SR}^{(0)} \left( g_{LR}^{(0)} \right)^2 \left[ 2(n-2) \mathcal{I}_1(\epsilon_{SR}, \epsilon_{LR}) + \mathcal{I}_2(\epsilon_{SR}, \epsilon_{LR}) \right]. \tag{A.37}$$

The contributions to $g_{LR}$ comes from two expansion terms. The first is:

$$\frac{g_{LR}^{(0)} \left( g_{SR}^{(0)} \right)^2}{2} \sum_{\alpha=1}^{n} \sum_{\substack{\eta,\rho=1 \\ \eta \neq \rho}}^{n} \sum_{\substack{\beta,\nu=1 \\ \beta \neq \nu}}^{n} \int d^2x \int_{|y|<r} d^2y \int_{|z|<r} d^2z \, \sigma(x) \varepsilon^{(\alpha)}(x) \varepsilon^{(\rho)}(y) \times \varepsilon^{(\eta)}(y) \varepsilon^{(\nu)}(z) \varepsilon^{(\beta)}(z). \tag{A.38}$$

In the above sums, one has terms in which there are two pairs of energy fields in the same replica, for instance when $\alpha = \rho \neq \nu$, and $\eta = \beta$. These terms are given by the $\mathcal{I}_1$ integrals where the average on the $\sigma$ are replaced by the average of $\varepsilon$. The result of this integral is given in Eq. (A.26) by replacing $\epsilon_{LR} = \epsilon_{SR}$. They produces $8 \times (n-1)(n-2)$ interaction terms $\sum_{\alpha=1}^{n} \sigma \varepsilon^{(\alpha)}$. Then one has the terms in Eq. (A.38) where there are three energy of the same replica at the three different points. These terms can be reduced to integral of type $\mathcal{I}_2$, again with $\mathbb{E}[\sigma\sigma]$ replaced by $\langle \varepsilon\varepsilon \rangle$ in Eq. (A.21). They produce $4 \times (n-1)$ interaction terms $\sum_{\alpha=1}^{n} \sigma \varepsilon^{(\alpha)}$. So the term Eq. (A.38) gives contribution $\mathcal{I}_{g_{SR}^2 g_{LR}}$ to $g_{LR}$:

$$\mathcal{I}_{g_{SR}^2 g_{LR}} = g_{LR}^{(0)} \left( g_{SR}^{(0)} \right)^2 \left[ 4(n-1)(n-2) \mathcal{I}_1(\epsilon_{SR}, \epsilon_{SR}) + 2(n-1) \mathcal{I}_2(\epsilon_{SR}, \epsilon_{SR}) \right]. \tag{A.39}$$

Finally we have the expansion term:

$$\frac{\left( g_{LR}^{(0)} \right)^3}{6} \sum_{\alpha=1}^{n} \sum_{\beta=1}^{n} \sum_{\nu=1}^{n} \int d^2x \int_{|y|<r} d^2y \int_{|z|<r} d^2z \, \sigma(x) \varepsilon^{(\alpha)}(x) \sigma(y) \varepsilon^{(\beta)}(y) \sigma(z) \varepsilon^{(\nu)}(z). \tag{A.40}$$

Among the terms in the sum, we have to distinguish the ones with two energy fields in the same replica, generating $\mathcal{J}_1$ contributions Eq. (A.22) and the ones with three energy fields in the same replica, associated to the $\mathcal{J}_2$ integrals in Eq. (A.23). Counting the corresponding combinatorial factors, one finds that Eq. (A.40) contributes with:

$$\mathcal{I}_{g_{LR}^3} = \left( g_{LR}^{(0)} \right)^3 \left[ \frac{n-1}{2} \mathcal{J}_1 + \frac{1}{6} \mathcal{J}_2 \right]. \tag{A.41}$$

Resuming we have:

$$g_{SR} = r^{\epsilon_{SR}} g_{SR}^{(0)} + \mathcal{I}_{g_{SR}^2} + \mathcal{I}_{g_{LR}^2} + \mathcal{I}_{g_{SR}^3} + \mathcal{I}_{g_{SR} g_{LR}^2}, \tag{A.42}$$

$$g_{LR} = r^{\epsilon_{LR}} g_{LR}^{(0)} + \mathcal{I}_{g_{SR} g_{LR}} + \mathcal{I}_{g_{LR}^3} + \mathcal{I}_{g_{SR}^2 g_{LR}}. \tag{A.43}$$

Among all the above terms, only the $\mathcal{I}_{g_{LR}^3}$ remains undetermined. Indeed, to compute its $(\epsilon_{LR}, \epsilon_{SR})$ expansion one needs to specify the $\sigma-$four point functions. We computed $\mathcal{I}_{g_{LR}^3}$ for two cases, a Gaussian distribution and an instance of non Gaussian one, see Appendix (D). In these two cases, we have observed that the renormalizability of the theory, in particular the fact that the divergences of type $\epsilon_{SR}^{-2}$ or $\epsilon_{LR}^{-2}$ are absorbed in the definition of the renormalized couplings, is broken for $n \neq 0$ by the above term. The renormalizability is however re-established in the limit $n \to 0$. This phenomena is also discussed in [3]. Analogously to their observations, we expect that there should be additional counterterms, coming for instance from the relevant fields in Eq.(4), that cancel in the limit $n \to 0$. As we are ultimately interested in the

disordered limit $n \to 0$, we can content ourselves with the renormalizability in this limit. This is satisfied when the above term takes the form:

$$\lim_{n \to 0} \mathcal{I}_{g_{LR}^3} = \pi^2 \frac{r^{2\epsilon_{LR}}}{2\epsilon_{LR}} \left[ \frac{-4}{2\epsilon_{LR} - \epsilon_{SR}} + \mathcal{C}\left(s = \frac{\epsilon_{SR}}{\epsilon_{LR}}\right) \right] + O(1), \tag{A.44}$$

where $\mathcal{C}(s)$ will be some function of the ratio $s = \epsilon_{SR}/\epsilon_{LR}$. As mentioned above, for the two distributions considered here we have indeed verified Eq. (A.44). In particular, for the Gaussian disorder distribution, $\mathcal{C}(s)$ is given by Eq. (D.6) and for the non-Gaussian disorder defined in Appendix (D.2), by Eq. (D.10).

Collecting all the above results, at the 2-loop order and in the replica limit $n \to 0$, we find the following renormalization for the couplings:

$$g_{SR} = r^{\epsilon_{SR}} \left( g_{SR}^0 - 8\pi \left(g_{SR}^0\right)^2 \frac{r^{\epsilon_{SR}}}{\epsilon_{SR}} + \pi \left(g_{LR}^0\right)^2 \frac{r^{2\epsilon_{LR} - \epsilon_{SR}}}{2\epsilon_{LR} - \epsilon_{SR}} + \pi^2 \left(g_{SR}^{(0)}\right)^3 \frac{r^{2\epsilon_{SR}}}{2\epsilon_{SR}} \left[ \frac{128}{\epsilon_{SR}} + 32 \right] \right.$$
$$\left. + \pi^2 \left(g_{LR}^{(0)}\right)^2 g_{SR}^{(0)} \frac{r^{2\epsilon_{LR}}}{2\epsilon_{LR}} \left[ \frac{-32\,\epsilon_{LR}}{\epsilon_{SR}(2\epsilon_{LR} - \epsilon_{SR})} + \frac{8}{\epsilon_{SR}} + \mathcal{B}(s) \right] \right), \tag{A.45}$$

$$g_{LR} = r^{\epsilon_{LR}} \left( g_{LR}^0 - 4\pi g_{LR}^0 g_{SR}^0 \frac{r^{\epsilon_{SR}}}{\epsilon_{SR}} + 8\pi^2 \left(g_{SR}^{(0)}\right)^2 g_{LR}^0 \frac{r^{2\epsilon_{SR}}}{2\epsilon_{SR}} \left[ \frac{6}{\epsilon_{SR}} + 1 \right] \right.$$
$$\left. + \pi^2 \left(g_{LR}^0\right)^3 \frac{r^{2\epsilon_{LR}}}{2\epsilon_{LR}} \left[ \frac{-4}{2\epsilon_{LR} - \epsilon_{SR}} + \mathcal{C}(s) \right] + O(g^4) \right), \tag{A.46}$$

where we recall again $s = \epsilon_{SR}/\epsilon_{LR}$.

# B  Zamolodchikov $c$-theorem

To compute the central charge, we use the Zamolodchikov $c$-theorem [47], according to which we have to find the function $C\left(g_{SR}, g_{LR}\right)$ of the couplings such that:

$$\beta_{SR} \frac{\partial}{\partial g_{SR}} C\left(g_{SR}, g_{LR}\right) + \beta_{LR} \frac{\partial}{\partial g_{LR}} C\left(g_{SR}, g_{LR}\right) = -6\pi^2 \left\langle \Theta(0)\Theta(1) \right\rangle, \tag{B.1}$$

where $\Theta(x)$ is the trace of the stress-energy tensor. This latter is related, by the renormalizability of the theory, to the perturbation terms by:

$$\Theta(x) = \beta_{SR} \sum_{\alpha \neq \beta}^n \varepsilon^{(a)}(x)\varepsilon^{(b)}(x) + \beta_{LR} \sum_a \sigma(x)\varepsilon^{(a)}(x). \tag{B.2}$$

Using:

$$\left\langle \sum_{a \neq b}^n \varepsilon^{(a)}(0)\varepsilon^{(b)}(0) \sum_{c \neq d}^n \varepsilon^{(a)}(1)\varepsilon^{(b)}(1) \right\rangle = 2\,n(n-1), \tag{B.3}$$

$$\left\langle \sum_a^n \sigma(0)\varepsilon^{(a)}(0) \sum_b^n \sigma(1)\varepsilon^{(b)}(1) \right\rangle = n, \tag{B.4}$$

$$\left\langle \sum_{a \neq b}^n \varepsilon^{(a)}(0)\varepsilon^{(b)}(0) \sum_c^n \sigma(1)\varepsilon^{(c)}(1) \right\rangle = 0, \tag{B.5}$$

one has:

$$\left\langle \Theta(0)\Theta(1) \right\rangle = 2n(n-1)\beta_{SR}^2 + n\beta_{LR}^2. \tag{B.6}$$

To compare the central charge with Monte Carlo results, the 1-loop order is enough. At this order, the $\beta$ functions, for the system with $n$-replicas, are:

$$\beta_{SR} = \epsilon_{SR}\, g_{SR} + 4\pi(n-2)g_{SR}^2 + \pi g_{LR}^2,$$
$$\beta_{LR} = \epsilon_{LR}\, g_{LR} + 4\pi(n-1)g_{SR}g_{LR}. \tag{B.7}$$

Using the above equations, one can verify that a solution of Eq. (B.1) is:

$$C\left(g_{SR}, g_{LR}\right) = C\left(0,0\right) - 6\pi^2 n \Bigg( (n-1)\epsilon_{SR}\, g_{SR}^2 + \frac{1}{2}\epsilon_{LR}\, g_{LR}^2 +$$
$$+ \frac{8\pi}{3}(n-2)(n-1)g_{SR}^3 + 2\pi(n-1)g_{LR}^2\, g_{SR} \Bigg). \tag{B.8}$$

The effective central charge $c_{eff}^X$ at the $X$ fixed point, $X = \{SR, LR\}$, is obtained by:

$$c_{eff}^X = \lim_{n\to 0}\frac{1}{n}\, C\left(g_{SR}^{*,X}, g_{LR}^{*,X}\right). \tag{B.9}$$

The central charge of the pure $q-$Potts model, see Eq. (36), fixes the initial condition $C\left(0,0\right) = c^P$. Using Eq. (B.8), one obtains the Eqs. (38) and (39).

## C  Renormalization of the energy field

We give here some detail on how we derive Eq. (45). We apply the same procedure explained in [15] to renormalize a field $O$, which schematically can be summarized as:

$$\mathcal{O}\left(1 + g\int \Phi^{\text{pert}}(x) + \frac{g^2}{2}\int \Phi^{\text{pert}}(x)\int \Phi^{\text{pert}}(y) + \cdots\right) \to Z_O \mathcal{O}, \tag{C.1}$$

using a generic $\Phi$ field. We consider the field Eq. (42). At first order we have:

$$g_{SR}^{(0)} \sum_{\alpha=1}^{n} \sum_{\substack{\eta,\rho=1 \\ \eta\neq\rho}}^{n} \int_{|y|<r} d^2 y\, \varepsilon^{(\alpha)}(0)\varepsilon^{(\rho)}(y)\varepsilon^{(\eta)}(y), \tag{C.2}$$

which, in case $\alpha = \rho$ or $\alpha = \eta$, contributes to $Z_\varepsilon$. This term was also studied in [15]. By simple counting, and using the energy-energy fields expansion, see Eq. (A.35), one has:

$$g_{SR}^{(0)}\, 2(n-1) \times \int_{|y|<r} d^2 y\, |y|^{-2h_\varepsilon} = 4\pi(n-1)\, g_{SR}^{(0)}\, \frac{r^{\epsilon_{SR}}}{\epsilon_{SR}}. \tag{C.3}$$

In the limit $n \to 0$, it is the first order correction seen in Eq. (45). Among the second order expansion terms, we have the one which was already analyzed in [15]:

$$\frac{\left(g_{SR}^{(0)}\right)^2}{2} \sum_{\alpha=1}^{n} \sum_{\substack{\eta,\rho=1 \\ \eta\neq\rho}}^{n} \sum_{\substack{\beta,\nu=1 \\ \beta\neq\nu}}^{n} \int_{|y|<r} d^2 y \int_{|z|<r} d^2 z\, \varepsilon^{(\alpha)}(0)\varepsilon^{(\rho)}(y)\varepsilon^{(\eta)}(y)\varepsilon^{(\nu)}(z)\varepsilon^{(\beta)}(z). \tag{C.4}$$

Here there are the terms with two pairs of energy fields in the same replica, for instance when $\alpha = \eta$ and $\rho = \eta$, $\alpha \neq \beta$. Again, from the short-distance expansion of the energy field, these contributions are given by the integral $\mathcal{I}_1$ by replacing $\mathbb{E}[\sigma\sigma] \to \langle\varepsilon\varepsilon\rangle$. By simple counting, one gets:

$$2(n-1)(n-2)\left(g_{SR}^{(0)}\right)^2 \mathcal{I}_1(\epsilon_{SR}, \epsilon_{SR}) = 16\pi^2(n-1)(n-2)\frac{r^{2\epsilon_{SR}}}{\epsilon_{SR}^2}. \tag{C.5}$$

Then we have the terms in which there are three energy fields in the same replica. These contribute with amplitudes $\mathcal{I}_2$ in Eq. (A.21) with $\mathbb{E}[\sigma\sigma] \to \langle\varepsilon\varepsilon\rangle$. One has:

$$2(n-1)\left(g_{SR}^{(0)}\right)^2 \mathcal{I}_2(\epsilon_{SR}, \epsilon_{SR}) = 2(n-1)\pi^2\left(g_{SR}^{(0)}\right)^2\left[\frac{8}{\epsilon_{SR}} - 4\right]\frac{r^{2\epsilon_{SR}}}{2\epsilon_{SR}}, \qquad \text{(C.6)}$$

where we used $\mathcal{B} = -4$ when $\epsilon_{SR} = \epsilon_{LR}$, see Eq. (20). Collecting the terms Eq. (C.5) and Eq. (C.6), one finds in the limit $n \to 0$ the $\left(g_{SR}^{(0)}\right)^2$ contribution given in Eq. (45). The new expansion term, comparing to the work of [15], is:

$$\frac{\left(g_{LR}^{(0)}\right)^2}{2}\sum_{\rho=1}^{n}\sum_{\eta=1}^{n}\int_{|y|<r}d^2y\int_{|z|<r}d^2z\ \varepsilon^{(\alpha)}(0)\sigma(y)\varepsilon^{(\rho)}(y)\sigma(z)\varepsilon^{(\eta)}(z). \qquad \text{(C.7)}$$

Here we have the terms in which there is only one pair of energies at the same replica, for instance $\alpha = \rho \neq \eta$, whose contribution is associated to $\mathcal{I}_1$:

$$(n-1)\left(g_{LR}^{(0)}\right)^2 \mathcal{I}_1(\epsilon_{SR}, \epsilon_{LR}) = \pi^2(n-1)\left(g_{LR}^{(0)}\right)^2\left[\frac{8\,\epsilon_{LR}}{\epsilon_{SR}(2\epsilon_{LR} - \epsilon_{SR})}\right]\frac{r^{2\epsilon_{LR}}}{2\epsilon_{LR}}, \qquad \text{(C.8)}$$

and the terms, associated to $\mathcal{I}_2$ where three energies have the same replica index:

$$\frac{\left(g_{LR}^{(0)}\right)^2}{2}\mathcal{I}_2(\epsilon_{SR}, \epsilon_{LR}) = \pi^2\left(g_{LR}^{(0)}\right)^2\left[\frac{4}{\epsilon_{SR}} + \frac{\mathcal{B}}{2}\right]\frac{r^{2\epsilon_{LR}}}{2\epsilon_{LR}}. \qquad \text{(C.9)}$$

Adding Eq. (C.8) and Eq. (C.9), one finds the $g_{LR}^2$ term in Eq. (45) in the $n \to 0$ limit.

# D  Two different disorder distributions

## D.1  Gaussian disorder distribution

Let us consider a Gaussian disorder distribution that satisfies Eq. (2). As in [3], the corresponding action $\mathcal{S}^{\text{aux}}$ is represented in terms of the non-local laplacian $\left(-\partial^2\right)^{\frac{2-a}{2}}$ [48,49]. In particular we take:

$$\sigma(x) = \sigma_G(x), \qquad \mathcal{S}^{\text{aux}} = \mathcal{S}^{\text{aux}}_{\text{Gauss}} = \int d^2x\ \sigma_G\left(-\partial^2\right)^{\frac{2-a}{2}}\sigma_G. \qquad \text{(D.1)}$$

For general values of $a$, the above action is non local and has only global conformal symmetry. The fusion $\sigma\sigma$ satisfies Eq.(4), with the composite field $\sigma^2$ appearing as a relevant field. We can then apply our RG approach knowing that, as explained in Section (2.2), this field does not affect the RG equations in the disorder limit $n \to 0$. The fact that $\sigma^2$ does not contribute in the disorder limit was already observed in [3].

From Wick theorem, the four-point function is easily computed:

$$\begin{aligned}
\mathbb{E}[\sigma_G(x)\sigma_G(y)\sigma_G(z)\sigma_G(w)] &= \mathbb{E}[\sigma_G(x)\sigma_G(y)]\mathbb{E}[\sigma_G(z)\sigma_G(w)]\\
&+ \mathbb{E}[\sigma_G(x)\sigma_G(z)]\mathbb{E}[\sigma_G(y)\sigma_G(w)]\\
&+ \mathbb{E}[\sigma_G(x)\sigma_G(w)]\mathbb{E}[\sigma_G(y)\sigma_G(z)].
\end{aligned} \qquad \text{(D.2)}$$

By placing a disorder field at infinite, the above expression becomes:

$$\mathbb{E}[\sigma_G(0)\sigma_G(y)\sigma_G(z)\sigma_G(\infty)] = \mathbb{E}[\sigma_G(o)\sigma_G(y)] + \mathbb{E}[\sigma_G(0)\sigma_G(y)] + \mathbb{E}[\sigma_G(y)\sigma_G(z)]. \qquad \text{(D.3)}$$

We can now use the Eq. (D.3) in the Eq. (A.22) and in the Eq. (D.3), finding that, for the leading $\epsilon_{LR}$ and $\epsilon_{SR}$ expansions:

$$\mathcal{J}_1 = 2\,\mathcal{I}_1\,, \qquad \mathcal{J}_2 = 3\,\mathcal{I}_2\,. \tag{D.4}$$

To obtain the first relation in the above equation, we use the fact that integrals such as :

$$\int_{|y|<r} d^2\,y \int_{|z|<r} d^2\,z\,|z-y|^{-2h_\varepsilon-2h_\sigma} = O(\epsilon^0)\,, \tag{D.5}$$

do not contain any singularities. Using Eq. (A.26) and Eq. (A.29), and comparing with Eq. (A.44), we obtain that:

$$\mathcal{C} \to \mathcal{C}^{\mathrm{G}} = \frac{\mathcal{B}}{2}\,. \tag{D.6}$$

## D.2  A non-Gaussian distribution

We can also compute $\mathcal{C}$ for a particular instance of a non-Gaussian disorder. Let us assume that $\sigma$ coincides with the Potts energy field, different from the other $n$ copies:

$$\sigma = \varepsilon\,, \qquad a = 2 - \epsilon_{SR}\,, \qquad \epsilon_{LR} = \epsilon_{SR}\,,$$
$$\mathbb{E}\left[\sigma(0)\sigma(y)\sigma(z)\sigma(\infty)\right] = \langle\varepsilon(0)\varepsilon(y)\varepsilon(z)\varepsilon(\infty)\rangle\,. \tag{D.7}$$

We point out that this disorder is not the one implemented in [1]. In this latter case indeed, there are more complicated integrals at the 2-loops order, whose computation cannot be done with the analytic techniques applied here. The choice (D.7) instead allows for an exact computation of the second-order loop contributions, thus showing explicitly, for the first time, the effects of higher cumulants. Moreover, the disorder (D.7) can be implemented in numerical simulations, making this choice not completely abstract.

In this case, the integrals $\mathcal{J}_1$ and $\mathcal{J}_2$ take the same form as the ones already considered in [15]. One obtains:

$$\mathcal{J}_1 = \mathcal{I}_2\,, \qquad \mathcal{J}_2 = 0\,. \tag{D.8}$$

As explained in [15], the second term in the above equation has been put to zero because the integral:

$$\int_{|y|<r} d^2\,y \int_{|z|<r} d^2\,z\,(\langle\varepsilon(0)\varepsilon(y)\varepsilon(z)\varepsilon(\infty)\rangle)^2 = O\left(\epsilon^0\right)\,, \tag{D.9}$$

has no singularities. Again, comparing with Eq. (A.44), one has

$$\mathcal{C} \to \mathcal{C}^{NG} = -\frac{\mathcal{B}}{2} = 2\,, \tag{D.10}$$

where we used the fact that $\epsilon_{SR} = \epsilon_{LR}$ for this case, see Eq. (D.7). We can see that, in this case, the relation $\nu^{LR} = 2/a$ is broken at 2-loop approximation:

$$\frac{1}{\nu^{LR}} = \frac{a}{2} - (2-a)^2 + O((2-a)^3)\,. \tag{D.11}$$

## E  Stability of the fixed points

We are interested in a $\epsilon^2$−order expansion of the stability matrix eigenvalues Eq. (24). We will focus here on the LR stability matrix (stability matrix computed at the LR point), since the

computation is more cumbersome with respect to the SR case. We can approach the problem using a first order perturbation scheme. We write:

$$M = M^{(0)} + M^{(1)}, \tag{E.1}$$

where $M^0$ has $O(\epsilon)$ element:

$$M^{(0)} = \begin{pmatrix} -4\epsilon_{LR} + \epsilon_{SR} & \sqrt{2 - \frac{\epsilon_{SR}}{\epsilon_{LR}}}\,\epsilon_{LR} \\ -2\sqrt{2 - \frac{\epsilon_{SR}}{\epsilon_{LR}}}\,\epsilon_{LR} & 0 \end{pmatrix}, \tag{E.2}$$

while $M^{(1)}$ has $O(\epsilon^2)$ entries:

$$M^{(1)} = \begin{pmatrix} \epsilon_{LR}^2\left(4 - 2\mathcal{C} + \frac{\mathcal{B}}{2}\right) + \epsilon_{LR}\epsilon_{SR}\left(\mathcal{C} - \frac{\mathcal{B}}{4}\right) & \frac{\epsilon_{LR}^2(8\mathcal{C} + 2\mathcal{B}) + \epsilon_{SR}\epsilon_{LR}(-2 - 6\mathcal{C} - \mathcal{B}) + \mathcal{C}\epsilon_{SR}^2}{8\sqrt{2 - \frac{\epsilon_{SR}}{\epsilon_{LR}}}} \\ \frac{\epsilon_{LR}^2(16 - 8\mathcal{C} + 2\mathcal{B}) + \epsilon_{SR}\epsilon_{LR}(-6 + 6\mathcal{C} - \mathcal{B}) - \mathcal{C}\epsilon_{SR}^2}{4\sqrt{2 - \frac{\epsilon_{SR}}{\epsilon_{LR}}}} & \mathcal{C}\left(\epsilon_{LR}^2 - \frac{\epsilon_{SR}\epsilon_{LR}}{2}\right) \end{pmatrix}. \tag{E.3}$$

We want to find the correction:

$$\lambda_1 = \lambda_1^{(0)} + \lambda_1^{(1)}, \qquad \lambda_2 = \lambda_2^{(0)} + \lambda_2^{(1)}, \tag{E.4}$$

where $\lambda_{1,2}^{(0)}$ are the eigenvalues of the "unperturbed" matrix $M_0$ (of order $\epsilon$) and the $\lambda_{1,2}^{(1)}$ are the first order correction (so of order $\epsilon^2$). When $\epsilon_{SR} > 0$ the matrix $M^{(0)}$ is diagonalizable. One has:

$$A^{-1}M^0 A = \begin{pmatrix} -2\epsilon_{LR} & 0 \\ 0 & -2\epsilon_{LR} + \epsilon_{SR} \end{pmatrix}, \tag{E.5}$$

where:

$$A = \begin{pmatrix} \frac{1}{\sqrt{2 - \frac{\epsilon_{SR}}{\epsilon_{LR}}}} & \frac{\sqrt{2 - \frac{\epsilon_{SR}}{\epsilon_{LR}}}}{2} \\ 1 & 1 \end{pmatrix}. \tag{E.6}$$

The zero order (corresponding to 1−loop order in the RG computation) eigenvalues are therefore:

$$\lambda_1^{(0)} = -2\epsilon_{LR}, \qquad \lambda_2^{(0)} = -2\epsilon_{LR} + \epsilon_{SR}, \tag{E.7}$$

and the corresponding eigenvectors:

$$\left|\lambda_1^{(0)}\right\rangle = \begin{pmatrix} \frac{1}{\sqrt{2 - \frac{\epsilon_{SR}}{\epsilon_{LR}}}} \\ 1 \end{pmatrix}, \qquad \left|\lambda_2^{(0)}\right\rangle = \begin{pmatrix} \frac{\sqrt{2 - \frac{\epsilon_{SR}}{\epsilon_{LR}}}}{2} \\ 1 \end{pmatrix}. \tag{E.8}$$

Note that when $\epsilon_{SR} = 0$ the eigenvalues of $M^{(0)}$ is double degenerate but the rank of $A$ is one (and not invertible). The matrix $M^{(0)}$ is an example of a so called defective matrix, whose eigenvectors span a space smaller than its dimension (in this case 2).

The $\lambda_{1,2}^{(1)}$ are found by:

$$\lambda_1^{(1)} = \left\langle \lambda_1^0 \middle| M^{(1)} \middle| \lambda_1^0 \right\rangle, \qquad \lambda_2^{(1)} = \left\langle \lambda_2^0 \middle| M^{(1)} \middle| \lambda_2^0 \right\rangle, \tag{E.9}$$

which is equivalent to:

$$\lambda_1^{(1)} = \left(A^{-1}M^{(1)}A\right)_{11}, \qquad \lambda_2^{(1)} = \left(A^{-1}M^1 A\right)_{22}. \tag{E.10}$$

One finds the results in Eq. (32).

Now it is quite manifest that one cannot obtain the result of Ising by setting $\epsilon_{SR} = 0$ in the previous results: indeed one finds a singularity, which can be traced back to the fact that the matrix $A$ is not invertible. One has to take the limit more carefully.

It is much more simple to diagonalize the matrix $M$ with $\epsilon_{SR} = 0$:

$$M(\epsilon_{SR} = 0) = \begin{pmatrix} -4\epsilon_{LR} + \epsilon_{LR}^2\left(4 - 2\mathcal{C} + \frac{\mathcal{B}}{2}\right) & \sqrt{2}\,\epsilon_{LR} + \epsilon_{LR}^2\left(\frac{\mathcal{C}}{\sqrt{2}} + \frac{\mathcal{B}}{2\sqrt{2}}\right) \\ -2\sqrt{2}\,\epsilon_{LR} + \epsilon_{LR}^2\left(2\sqrt{2} - \sqrt{2}\mathcal{C} + \frac{\mathcal{B}}{2\sqrt{2}}\right) & \mathcal{C}\epsilon_{LR}^2 \end{pmatrix}, \quad \text{(E.11)}$$

obtaining the Eq. (28). Now the corrections to $M^{(0)}$ shift the degenerations of the eigenvalues making $M$ diagonalizable. This is done by developing a pair of complex eigenvalues, one conjugate with the other.

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
