# Peer review of "Two-dimensional Ising and Potts model with long-range bond disorder: a renormalization group approach"

_SciPost Physics, doi:SciPost Phys. 15, 135 (2023)_

## Round 1 · Referee Report · Anonymous (Referee 1) · 2023-7-10

Strengths

1- First RG calculation (exploiting conformal invariance) of the Potts model with long-range disorder. 2- Confirms the existence of a stability lower bound $a^*$ of the Long-Range Fixed Point conjectured from numerical simulations 3- Confirms the violation of the Weinrib-Halperin law $\nu=2/a$ for non-Gaussian disorder.

Weaknesses

1- Expansion around $a=2$, far from the interesting region $a^*\simeq 0.75$.

Report

The authors studied the 2D $q$-state Potts model with long-range disorder by means of a real-space Renormalization Group technique. Both the energy density of the pure model and the disorder field are assumed to be primary fields of some Conformal Field Theories, which constrains the form of their correlation functions. The results, location of the RG fixed points and critical exponents, are expressed in terms of a double expansion with $q-2$ and $2-a$ where $a$ is the decay exponent of disorder correlations. The same model was very recently studied by the authors by Monte Carlo simulations.

The study of the stability of the Short-Range and Long-Range Fixed points shows that the change of stability of these fixed points occurs when $a=2/\nu_{\rm SR}$, as predicted by the Harris criterion. The Long-Range Fixed Point remains stable down to $a^*$. The predicted numerical values
of $a^*$ (0 for the Ising model and $1.6$ for the 3-state Potts) are far from the Monte Carlo estimates $a^*\simeq 0.75$. As emphasized by the authors, one cannot expect an expansion with $2-a$ to be valid up to $a=0.75$. Nevertheless, it is interesting that the RG calculation confirms the existence of a lower stability bound $a^*$, as observed numerically.

The central charge was estimated at the two disordered Fixed Points. The agreement with the Monte Carlo data (Figure 2) is very impressive. It is extremely interesting that the central charge of the long-range Fixed Point becomes compatible with that of the Infinite-Disorder Fixed Point precisely at the numerical estimate $a^*\simeq 0.75$. However, I am wondering what credence to give to this since, again, the central charge is obtained as an expansion in $2-a$ small. Could the authors comment on that?

Finally, the correlation length exponent $\nu$ is computed at the two disordered Fixed Points. At the Long-Range Fixed Point, $\nu$ is compatible with the Weinrib-Halperin conjecture $\nu=2/a$ wih Gaussian disorder. However, a violation is observed for non-Gaussian disorder. Again, this is something that was suspected from numerical simulations and that is very nicely confirmed by RG calculations.

In conclusion, several new interesting results are provided by this study that undoubtedly deserves to be publish in Scipost. The paper is clearly written and, I believe, accessible to non-experts. The appendices are very useful to understand/learn about the technical details of the calculations.

Requested changes

I have found a few minor typos: 1- It seems to me that a factor $\pi$ is missing in the denominator of the r.h.s. of equation (2.13), 2- remormalized --> renormalized at the bottom of page 16, 3- with an an --> with an (before Eq. A.8) 4- $d^2$ --> $d^2x$ in Eq. A.8 and A.11

  • validity: top
  • significance: top
  • originality: top
  • clarity: high
  • formatting: excellent
  • grammar: perfect

Author:  Francesco Chippari  on 2023-08-07  [id 3877]

(in reply to Report 1 on 2023-07-10)
Category:
answer to question
correction

First, we would like to thank the referee for their remarks and insightful observations that allowed us to improve the clarity of the paper.

1- It seems to me that a factor π is missing in the denominator of the r.h.s. of equation (2.13).

Answer:

The referee is right, we have corrected Eq (2.14) (in the new version). To be more clear, we have also specified in the new version after Eq.(2.14) the branch of the function arccos which is implied.

2- remormalized --> renormalized at the bottom of page 16, 3- with an an --> with an (before Eq. A.8) 4- d^2 --> d^2 x in Eq. A.8 and A.11

Answer:

Typos corrected.

Author:  Francesco Chippari  on 2023-08-07  [id 3876]

(in reply to Report 1 on 2023-07-10)
Category:
answer to question
correction

First, we would like to thank the referee for the remarks and insightful observations that allowed us to improve the clarity of the paper.

1- It seems to me that a factor π is missing in the denominator of the r.h.s. of equation (2.13).

Answer:

The referee is right, we have corrected Eq (2.14) (in the new version). To be more clear, we have also specified in the new version after Eq.(2.14) the branch of the function arccos which is implied.

2- remormalized --> renormalized at the bottom of page 16, 3- with an an --> with an (before Eq. A.8) 4- d^2 --> d^2 x in Eq. A.8 and A.11

Answer:

Typos corrected.

---

## Round 1 · Referee Report · Anonymous (Referee 2) · 2023-7-20

Strengths

1-Provides a perturbative RG-treatment of an interesting model of long-range quenched disorder.
2-Points out a limitation of the argument leading to the Weinrib-Halperin conjecture and provides a counterexample.
3-Provides a numerical test the prediction for the central charge in the Ising case.

Weaknesses

1-Does not provide a numerical check for the central charge in a case with q>2.

Report

The paper studies a very interesting model of quenched disorder, with two parameters $q$ and $a$. Following earlier work by the same authors, they here provide a two-loop perturbative CFT treatment in two dimensions, leading to substantial new results. I have a favorable opinion about this work, but I would like to see the following points discussed:
1-Early in the introduction a clear definition of $a$ should be given.
2-In connection with the discussion of the phase diagram, how does the LRp phase relate to the infinite-disorder fixed point found in cond-mat/9705038 and further discussed in cond-mat/9711279?
3-Around (4.8) can the authors argue or justify that there is no replica-symmetry breaking in this model?
4-Around (4.9) can the authors argue by symmetry or otherwise that the field $\epsilon$ cannot mix with $\sigma$, maybe using their transformation under duality?
5-It would be nice if the numerical check for the Ising model were extended to the $q$-Potts case with $q>2$, e.g. using techniques like in the references cited above.

Requested changes

See report.

  • validity: top
  • significance: high
  • originality: high
  • clarity: high
  • formatting: excellent
  • grammar: good

Author:  Francesco Chippari  on 2023-08-07  [id 3878]

(in reply to Report 2 on 2023-07-20)
Category:
answer to question
correction

First, we would like to thank the referee for their remarks and insightful observations that allowed us to improve the clarity of the paper.

1-Early in the introduction a clear definition of a should be given.

Answer:

We have added a ref to the Eq. 2.2 (of the new version), where a is precisely defined.

2-In connection with the discussion of the phase diagram, how does the LRp phase relate to the infinite-disorder fixed point found in cond-mat/9705038 and further discussed in cond-mat/9711279?

Answer:

We expect that the LRp phase is the same as the infinite-disorder fixed point of the above references in the uncorrelated disorder case. And indeed, the central charge shown in Figure 2 goes toward the expected value computed by Cardy and Jacobsen, c = 5 sqrt(3)/4/\pi*log(2) ~ 0.4777 in the large a limit. In fact, the value that we measured (shown as LRp in Fig.2) goes to a slightly lower value, which is due to the way we generate the correlated disorder.

We add a sentence (end of sec. 4.1) mentioning this point and referring to the two above mentioned papers. We also changed the notation LR (r=infty) by LRp in the Fig. 2.

3-Around (4.8) can the authors argue or justify that there is no replica-symmetry breaking in this model?

Answer:

The referee points out an important point. Indeed the assumption of a no-replica symmetry breaking has been made but in the old version it was not stated explicitly. Note that this is related also to the question 2 of Referee 3, see below.
Unfortunately, we have no convincing arguments that there is no replica-symmetry breaking, besides the fact this assumptions is believed to be valid for the short range disorder.
We have now made explicit this assumption in the new version, see the second paragraph of section 4. We have added the references [29],[30] and [31], which discuss this issue for the short range disorder, in the new version.

4-Around (4.9) can the authors argue by symmetry or otherwise that the field ϵ cannot mix with σ , maybe using their transformation under duality?

Answer:

The referee arises another important point. We do not see any symmetry argument to support the fact there is no mixing between the energy field and the disorder field. The mixing takes in general place for a finite number n of copies. So one has to search for a symmetry or a duality transformation which occurs only in the limit n->0. Note that, as showed in the references [32],[33] of the new version, the no-mixing of the sigma field comes from the non-locality of its measure.

5-It would be nice if the numerical check for the Ising model were extended to the q -Potts case with
q > 2 , e.g. using techniques like in the references cited above.

Answer:

The numerical check for the 3-Potts case is simple to do and we have done it and the agreement is also good.
But we do not want to present it here since it will require many explanations on the numerical details. Indeed, for q=3, there is, in addition to the long range fixed points, a disorder short range fixed point, for which the determination of the amount of disorder is non trivial. We will present these results in a subsequent work which will contain more numerical results.

We mention this point in the last sentence of sec. 4.1

---

## Round 1 · Referee Report · Slava Rychkov (Referee 3) · 2023-7-22

Report

This paper studies bond disordered Potts model with long range disorder. It relies on the replica approach. The disorder is modeled by a field $\sigma$ of scaling dimension $a/2$ which is close to 1. The paper focuses on the regime where $a<2$ is close to 2 and $q>2$ is close to 2. In this case they find a disordered perturbative fixed point and obtain predictions of critical exponents and the central charge in a double perturbative expansion in $2-a$ and $q-2$.

I should say that I am not an expert on the bond-disordered Potts model. I do not normally referee or edit papers from this subfield - I accepted out of curiosity.

Still, I have worked on long-range models and on disordered models in the past, and reading the paper I could formulate some hopefully not irrelevant questions.

1) The authors don't discuss how reasonable it is to model the long-range disorder by a CFT field. I kind of understand that a Gaussian long-range disorder, their example in App. D.1, may appear in some situations, but how reasonable it is to expect that a complicated disorder like in App. D.2 will arise? Is this physically likely or will this require some unreasonable finetuning? In Monte Carlo simulations Ref. [1] by Chiparri, Picco, Santachiara, with long-range disorder, was disorder more like D.1 or like D.2? It would be nice to know this, to assess the importance of their finding concerning the violation of the Weinrib-Halperin conjecture. Was that conjecture ever claimed to be true by its authors for non-Gaussian disordered?

2) The authors treat their disordered fixed point as if it were a local fixed point, e.g. computings its central charge. However I would suspect that for the Gaussian disorder distribution in App. D.1 the fixed point is non-local (i.e. it does not have a local conserved stress tensor). That's what happens when one couples a local CFT to a (non-local) Gaussian field. I studied a similar situation in 1703.03430,1703.05325 (the local Ising CFT in 2d and in 3d, perturbed by the coupling $\sigma \chi$ where $\sigma$ is the Ising field and $\chi$ a non-local Gaussian field) and I wonder if the authors see any connection or have any comment.

3) Minor comments:

  • Eq. (2.3) - it's not true for the example in App. D.1 where the relevant quasiprimary $\sigma^2$ appears in the r.h.s. It should be noted that the theory in D.1 has no Virasoro invariance, only global conformal invariance.

  • p.11 "the free energy is a self-averaging observable" - a reference or an argument would be welcome

  • I am not sure "since longtime" is a correct English usage

  • validity: -
  • significance: -
  • originality: -
  • clarity: -
  • formatting: -
  • grammar: -

Author:  Francesco Chippari  on 2023-08-07  [id 3879]

(in reply to Report 3 by Slava Rychkov on 2023-07-22)

First, we would like to thank the referee for their remarks and insightful observations that allowed us to improve the clarity of the paper.

1.a- The authors don't discuss how reasonable it is to model the long-range disorder by a CFT field. I kind of understand that a Gaussian long-range disorder, their example in App. D.1, may appear in some situations, but how reasonable it is to expect that a complicated disorder like in App. D.2 will arise? Is this physically likely or will this require some unreasonable finetuning?

Answer:

One should distinguish between experimental, theoretical and numerical works. In the real life realisations of long-range disordered critical systems, the disorder is in general not Gaussian, and its full distribution not known. For instance in the Ref 16 of the new version, the cumulants of the disorder distribution are given by the many points connectivities of a cluster obtained by a Diffusion-Limited-Aggregation (DLA) process, which are not known (moreover the fact the DLA processes satisfy Virasoro symmetry remains a very open question.)

Theoretical works (before ours) only considered Gaussian ones. The numerical works used Gaussian and non-Gaussian CFT disorder distributions. For some non-Gaussian distributions appearing in the literature, there is no evidence that they are described by a local CFT (even more that they can be described by a QFT framework). On the other hand, other numerical works found convenient way to generate non-Gaussian disorder by using critical lattice systems, described in the continuum limit by a local CFT. One example is the so-called "thermally diluted Ising model (see ref[19] of the new version), where the disorder is generated by considering an auxiliary pure 3D critical Ising model.

The disorder in App D.2 has not been numerically implemented so far, but it is easily doable. In the setting of our Ref [1], this is done by simulating a critical Potts model and taking sigma as the energy lattice Potts operator. The example of D.2 thus provides predictions that can be tested. The non-Gaussian disorder used in our Ref [1] requires more effort to compute the 2-loop diagrams. We are actually working on that.

1.b- In Monte Carlo simulations Ref. [1] by Chiparri, Picco, Santachiara, with long-range disorder, was disorder more like D.1 or like D.2? It would be nice to know this, to assess the importance of their finding concerning the violation of the Weinrib-Halperin conjecture.

Answer:

In Ref [1] we use the disorder D.1 and a disorder more like D.2, i.e. a Virasoro conformal distribution. The disorder D.1 was generated by using the fast Fourier transform method, in the same way therefore used by many other numerical simulations, see for instance refs Ballesteros-Parisi ([8]) or Kazmin-Janke([10],[11]). But many results presented in Ref[1] were done by taking sigma as the the product of the spin fields of copies of critical Ising theories.

The above answers are now integrated in a modified Section 2.1.

1.c- Was that conjecture ever claimed to be true by its authors for non-Gaussian disordered?

Answer:

Weinrib Halperin provided an argument that is not based on the fact the disorder is Gaussian (after some reflection, one can realise their argument can be made exact for a Gaussian disorder as done by Honkonen at al., ref [3]). As a matter of fact, many subsequent numerical works compared their results with the Weinrib Halperin conjecture, without making any comments on the fact their disorder was not Gaussian. We states this more clearly in the new version in Section 4.2, see also answer below.

2- The authors treat their disordered fixed point as if it were a local fixed point, e.g. computings its central charge. However I would suspect that for the Gaussian disorder distribution in App. D.1 the fixed point is non-local (i.e. it does not have a local conserved stress tensor). That's what happens when one couples a local CFT to a (non-local) Gaussian field. I studied a similar situation in 1703.03430,1703.05325 (the local Ising CFT in 2d and in 3d, perturbed by the coupling σχ where σ is the Ising field and χ a non-local Gaussian field) and I wonder if the authors see any connection or have any comment.

Answer:

We thank the referee and Emilio Trevisani (with whom we have had a long discussion after the submission of the paper) for having pointed us these references and the corresponding main results. In particular, Emilio pointed us the strict analogy between the so called shadow relations proven in these works and the Weinrib conjecture. We have modified the last paragraph of Section 4.2 where we comment on that. In particular, the crucial point is the fact that the non-locality of the Gaussian scale invariant theory implies the fact the Gaussian field is not renormalised. This was also explained in detail in the work of Honkonen Ref [3]. Concerning the use of the effective central charge, we have now made more clear the assumptions behind that. In particular, we have argued that, in the case the replicated theory is conformal, the question is related to the occurence of a replica symmetry breaking (see also the the questions of referee 2 above). Notice also that our predictions are compared to the numerical finding for the non-Gaussian disorder of Ref[1], where the replicated theory is local conformal.

Minor comments:

3.a- Eq. (2.3) - it's not true for the example in App. D.1 where the relevant quasiprimary σ^2 appears in the r.h.s. It should be noted that the theory in D.1 has no Virasoro invariance, only global conformal invariance.

Answer:

In the new version we have replaced the previous condition with a less constraining condition Eq (2.4). Indeed, our RG approach is still valid also in the presence of additional relevant fields, say \Phi, provided the composite field \Phi energy-energy is irrelevant (as it is the case for the Gaussian disorder). This is now explained in Section 2.2 (note in the new version we have added an equation, now Eq. 2.8). We have added some comments in Appendix D.1. In the new version we have also stressed throughout the paper the fact the theory D.1 has not local conformal symmetry.

3.b- p.11 "the free energy is a self-averaging observable" a reference or an argument would be welcome.

Answer:

For uncorrelated disorder, the free energy is expected to be a self-averaging observable, we add a reference discussing this point. For the correlated disorder that we consider in this work, we checked that free energy samples are normally distributed as a function of the disorder. In the new version, the discussion on self-averaging is moved few lines below, after we describe the measurement of the free energy on strips.

3.c- I am not sure "since longtime" is a correct English usage

Answer:

It has been corrected.

---

## Editorial Decision

published